# DIALOGUE AS DISCOVERY: NAVIGATING HUMAN INTENT THROUGH PRINCIPLED INQUIRY

## ABSTRACT

A fundamental bottleneck in human-AI collaboration is the "intention expression gap," the difficulty for humans to effectively convey complex, high-dimensional thoughts to AI. This challenge often traps users in inefficient trial-and-error loops and is exacerbated by the diverse expertise levels of users. We reframe this problem from passive instruction following to a Socratic collaboration paradigm, proposing an agent that actively probes for information to resolve its uncertainty about user intent. we name the proposed agent Nous, trained to acquire proficiency in this inquiry policy. The core mechanism of Nous is a training framework grounded in the first principles of information theory. Within this framework, we define the information gain from dialogue as an intrinsic reward signal, which is fundamentally equivalent to the reduction of Shannon entropy over a structured task space. This reward design enables us to avoid reliance on costly human preference annotations or external reward models. To validate our framework, we develop an automated simulation pipeline to generate a large-scale, preference-based dataset for the challenging task of scientific diagram generation. Comprehensive experiments, including ablations, subjective and objective evaluations, and tests across user expertise levels, demonstrate the effectiveness of our proposed framework. Nous achieves leading efficiency and output quality, while remaining robust to varying user expertise. Moreover, its design is domain-agnostic, and we show evidence of generalization beyond diagram generation. Experimental results prove that our work offers a principled, scalable, and adaptive paradigm for resolving uncertainty about user intent in complex human-AI collaboration.

## 1 INTRODUCTION

The transition of AI from an efficient tool to a true collaborative partner hinges on solving a core challenge: achieving a shared understanding with the user (Liang & Banks, 2025). While Large Language Models (LLMs) demonstrate remarkable fluency in text generation, their passive, instruction-following nature falters when faced with the inherent incompleteness of human intent expression (Shneiderman, 2022). This limitation is especially evident in creative and technical domains (Amershi et al., 2019; Fan et al., 2023). In such settings, users may hold highly innovative ideas yet struggle to articulate them with precision (Chang et al., 2025). When attempting to realize these ideas with AI, they often fall into a frustrating "guessing game," which in turn forces task goals to emerge gradually and be refined through collaborative processes (Oihane et al., 2024). The gap between a user's high-dimensional mental model and their ability to convey it in a machine-readable format has been described as the "intention gap," (Vanessa et al., 2024) which forces collaboration into inefficient trial-and-error loops (Buçinca et al., 2020). As a result, the entire burden of precise articulation falls on the human, and this paradigm is fundamentally unsustainable for complex tasks.

Our research stems from a core insight: Why must humans always painstakingly teach the AI, instead of the AI intelligently guiding the human? We advocate for a paradigm shift: envisioning AI not as a passive follower, but as an agent actively bridging this gap (McGrath et al., 2024; Haase & Pokutta, 2024). Inspired by the Socratic method, we treat it not merely as pedagogy but as a model for collaborative discovery (Liu et al., 2024). A Socratic agent does not simply await commands; it formulates strategic questions to systematically resolve its uncertainty about the user's goal (Patil & Patwardhan, 2020; Sahu, 2024). Each question-answer turn becomes a deliberate act of information

seeking, designed to maximize convergence toward a shared, high-fidelity understanding (Elmqvist et al., 2025; Yao et al., 2025; Khorsand & Pourahmadi, 2025; Thomas & Houssineau, 2024).

To this end, we introduce Nous, an agent designed to acquire proficiency in an optimal inquiry policy. The central mechanism of Nous is a training framework grounded in the first principles of information theory (Cover & Thomas, 2006; Wu et al., 2025; Khandelwal et al., 2025).Within this framework, we define the information gain from dialogue as an intrinsic reward, formally equivalent to the reduction of Shannon entropy over possible task specifications. By relying on this objective and computationally tractable signal,Nous avoids dependence on costly human preference annotations or external reward models (Spera & Agrawal, 2025; Li et al., 2023; Agarwal et al., 2022).

To validate this framework, we select scientific diagram generation as our testbed, a prototypical instance of the intention gap. The task is both high-dimensional and logically structured, providing objective criteria for evaluation while remaining sufficiently challenging (Basole & Major, 2024; Han et al., 2023). Building on this, we construct an automated simulation pipeline to generate a large-scale, preference-based dataset tailored to this setting (Shao et al., 2024). Finally, we conducted comprehensive experiments and evaluations, which demonstrated the effectiveness of our method. Moreover, the framework is domain-agnostic: we further show evidence of generalization beyond diagram generation through additional experiments in co-creative contexts (Haase & Pokutta, 2024; Singh et al., 2025). (1) **Nous**, an intelligent agent that instantiates the Socratic interaction paradigm with structured belief modeling. (2) **An information-theoretic reinforcement learning framework**, using dialogue-driven information gain as an intrinsic reward and eliminating the need for human annotation or external reward models. (3) **An automated large-scale simulation pipeline**, generating dialogue strategy learning data to support scalable training and evaluation.

## 2 RELATED WORK

Our work is situated at the intersection of three key areas in AI and human-computer interaction: goal-oriented dialogue, active learning, and large language model alignment.

**Goal-Oriented Dialogue Systems.** Traditional task-oriented dialogue (TOD) systems, typified by datasets like MultiWOZ (Budzianowski et al., 2018; Ramadan et al., 2018; Eric et al., 2019; Zang et al., 2020), excel in explicit slot-filling tasks such as booking flights (Young et al., 2013; Wen et al., 2017). However, these systems operate on a convergent retrieval" paradigm, assuming a fixed set of slots to retrieve a pre-existing database entry. In contrast, creative design tasks involve divergent construction," where the goal is to create a novel specification from scratch, requiring dynamic attribute combinations rather than static forms. While recent LLM-based approaches explore proactive clarification in QA (Lee et al., 2023b; Darji & Lutellier, 2025) or future-planning (Xu et al., 2024), most remain passive recipients of instructions. Our work moves beyond both traditional TOD and passive LLMs: Nous navigates a combinatorially complex specification space to resolve ambiguity, transforming the agent into an active inquirer for open-ended construction.

**Active Learning and Optimal Experiment Design.** The principle of reducing uncertainty by asking questions is rooted in active learning and optimal experiment design (Beluch et al., 2018; Lewis & Gale, 1994). Prior dialogue-policy research has incorporated entropy reduction as a signal for clarification (Padmakumar & Mooney, 2020), and recent studies formalize question quality directly via expected information gain (Mazzaccara et al., 2024; Geishauser et al., 2021; Xing et al., 2024). However, these methods typically target static datasets or constrained "20-questions" benchmarks. Our contribution is to extend this principle to dynamic dialogue for creative design: instead of selecting a data point, Nous learns to generate natural language questions that probe a latent goal space. Training this generative policy with entropy reduction as a real-time reward bridges classical theory with modern LLM interaction (Piriyakulkij et al., 2024; Chen et al., 2025; Zhao et al., 2025).

**LLM Alignment and Preference-Based Learning.** Aligning LLMs with human intent is a central challenge. Preference-based methods such as RLHF (Christiano et al., 2017; Ouyang & et al., 2022), PPO-based optimization (Schulman et al., 2017), and more recent approaches like GRPO (Shao et al., 2024), DPO (Rafailov et al., 2023), and RLAIF (Bai et al., 2022; Lee et al., 2023c) rely on costly preference labels or heuristic feedback. Our method offers a scalable alternative: we define an intrinsic reward from information gain, bypassing external reward models and the associated annotation cost. By applying offline RL (Levine et al., 2020; Kostrikov et al., 2021) on automatically

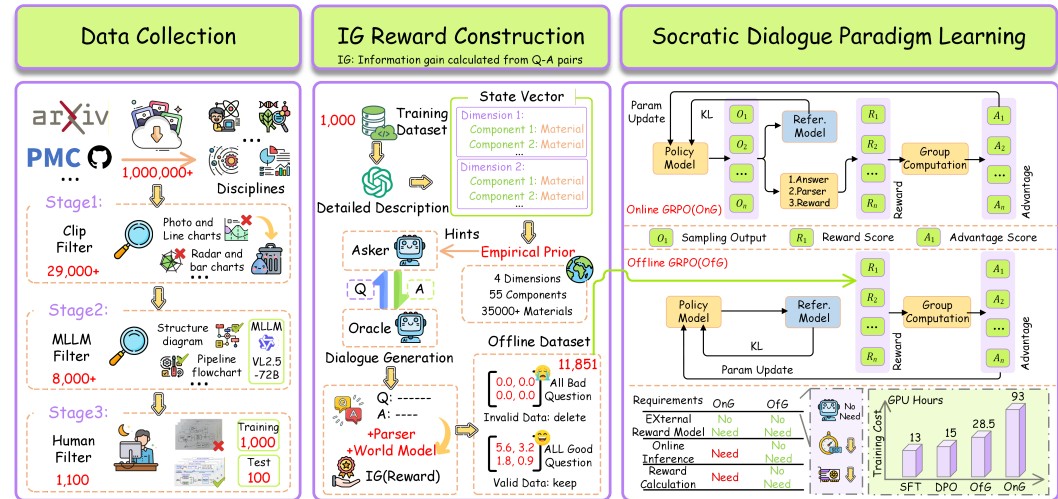

Figure 1: **The multi-stage curation pipeline for the dataset and the details of model training.**
We began with a raw dataset of approximately 1 million figures downloaded from scientific papers
in different fields on arXiv and PMC. This dataset was first filtered using the CLIP model to remove
data plots (such as bar charts and line graphs), resulting in 29,000 images. Next, we used the
Qwen-2.5-VL-72B model to retain true schematic diagrams, reducing the dataset to 8,000 images.
Finally, three PhD students conducted a manual review to ensure the relevance, clarity, and quality
of each figure, resulting in a final dataset of 1,100 images. From this curated dataset, 1,000 figures
were used to build the empirical prior and train simulations, while 100 figures were set aside for
testing. Detailed explanations regarding data distribution and open-source licenses are provided in
Appendix .

generated preferences, Nous avoids proxy misspecification while maintaining principled grounding
in task structure, offering a complementary path for alignment in structured co-creative tasks.

**AI for Design and Creativity.** A growing body of work envisions AI as a co-creative partner in
domains such as design and engineering (Tang et al., 2024; Singh et al., 2025). Most systems em-
phasize generation-providing suggestions or auto-completions. However, effective creation depends
on a well-defined goal. Our approach is unique in focusing on the "front-end" of co-creation: clar-
ifying the user's initial, ambiguous intent through dialogue. This emphasis on intent understanding
complements existing generative systems and lays a stronger foundation for accurate, relevant, and
user-aligned downstream outputs.

## 3 METHODOLOGY

Our methodology is presented in three parts. First, we establish a formal information-theoretic
framework, deriving an intrinsic and tractable reward signal from first principles (Sec. 3.1). Next, we
detail the complete offline training pipeline, which includes an automated simulation for preference
data generation and the offline policy optimization algorithm. (Sec. 3.2). Finally, we introduce the
baseline models used for our comparative experiments (Sec. 3.3).

### 3.1 AN INFORMATION-THEORETIC FRAMEWORK FOR OPTIMAL INQUIRY

To learn an effective inquiry strategy, the agent requires a quantitative metric for guidance. Drawing
from classical information theory, we define a reward signal based on information gain, which mea-
sures the informational value of each question-answer turn. We validate our method on the scientific
chart generation task, where the dialogue is modeled as a process of reducing epistemic uncertainty
over a structured state space. The information gain from a user's response is formally defined as the
Kullback-Leibler (KL) divergence between the posterior and prior belief states(the agent's probabil-
ity distribution over user intentions). We prove this metric simplifies to the reduction in the system's

Shannon entropy. This provides an intrinsic reward signal, directly calculable from the agent's belief state, for optimizing the inquiry policy without requiring a separate, pre-trained reward model.

**Formalizing the Diagram Specification Space.** We begin by defining the object of our inquiry. A complete scientific diagram specification, denoted by $\mathcal{G}$, is conceptualized as a point in a high-dimensional, discrete state space. A diagram specification is represented by a set of $N$ attributes, $\mathcal{V} = \{V_1, V_2, \ldots, V_N\}$. Each attribute $V_i$ takes a value $v_i$ from its finite, discrete domain $\mathcal{S}_i$. A complete and valid diagram specification is an instantiation $\mathbf{g} = (v_1, v_2, \ldots, v_N)$ where $v_i \in \mathcal{S}_i$ for all $i \in \{1, \ldots, N\}$. The attributes are designed to be comprehensive, covering aspects such as overall layout ($V_{\text{layout}}$), color palettes ($V_{\text{color}}$), the number and types of components ($V_{\text{num\_comp}}$, $V_{\text{comp\_type}}^{(k)}$), and interconnections ($V_{\text{conn}}^{(i,j)}$).

**Quantifying and Decomposing Epistemic Uncertainty.** At any turn $t$ in the dialogue, the agent's knowledge about the user's desired diagram is captured by a probabilistic belief state, $P_t(\mathcal{G})$. For computational tractability, we assume the attributes $V_i$ are conditionally independent given the dialogue history $\mathcal{H}_t$. While this is a simplifying assumption, we argue it is a tractable and effective first-order approximation, as the greatest reduction in uncertainty, particularly in early dialogue, comes from resolving major, orthogonal attributes (e.g., overall layout, number of components).

This allows the joint distribution to be factorized:

$$P_t(\mathcal{G}) = P(V_1, \ldots, V_N \mid \mathcal{H}_t) = \prod_{i=1}^{N} P(V_i \mid \mathcal{H}_t). \tag{1}$$

The agent's initial belief state, $P_0(\mathcal{G})$, is an empirical prior derived from a large-scale corpus $\mathcal{D}$ of existing diagrams, where each prior probability is estimated via maximum likelihood:

$$P_0(V_i = v_j) = \frac{\text{Count}_{\mathcal{D}}(V_i = v_j)}{|\mathcal{D}|}. \tag{2}$$

The total uncertainty of the system is the Shannon entropy of the belief state $P_t(\mathcal{G})$. A critical consequence of the independence assumption is that the total entropy decomposes into a sum of marginal entropies:

$$\mathcal{H}(P_t(\mathcal{G})) = -\sum_{\mathbf{g} \in \mathcal{G}} P_t(\mathbf{g}) \log_2 P_t(\mathbf{g}) = \sum_{i=1}^{N} \mathcal{H}(P_t(V_i)), \tag{3}$$

where $\mathcal{H}(P_t(V_i)) = -\sum_{v_j \in \mathcal{S}_i} P_t(V_i = v_j) \log_2 P_t(V_i = v_j)$. This decomposition is crucial, as it allows us to track uncertainty on a per-attribute basis.

**Belief State Update and Reward Function.** The dialogue proceeds as a sequence of belief state updates. An answer $A_t$ is mapped by a semantic parser $f$ to evidence $\mathcal{E}_t$, which imposes hard constraints on a subset of attributes $\mathcal{V}_{\mathcal{E}_t}$. In our simulation, $f$ is implemented as a few-shot prompted LLM, whose parsing accuracy is ensured by the Oracle's templated responses, providing a reliable signal for reward calculation. This updates the belief from a prior $P_t$ to a posterior $P_{t+1}$ via Bayesian conditioning. For any constrained attribute, the posterior becomes a deterministic Kronecker delta function, $P_{t+1}(V_i = v_j) = \delta_{jk}$, while unconstrained attributes remain unchanged.

We define our reward signal $r_t$ as the *reduction in Shannon entropy* of the belief state:

$$r_t \equiv IG(A_t) = \mathcal{H}(P_t(\mathcal{G})) - \mathcal{H}(P_{t+1}(\mathcal{G})). \tag{4}$$

Intuitively, this quantity measures the informational value of the user's answer. From an information-theoretic perspective, the expected value of this entropy reduction equals the mutual information between $A_t$ and $\mathcal{G}$, which can be written as an expectation over a KL divergence:

$$\mathbb{E}[IG(A_t)] = I(A_t; \mathcal{G}) = \mathbb{E}_{A_t}\big[ D_{KL}\big(P_{t+1}(\mathcal{G}) \,\|\, P_t(\mathcal{G})\big)\big]. \tag{5}$$

Thus maximizing information gain is identical to maximizing the reduction of uncertainty.

By substituting the entropy decomposition from Eq. 3 into Eq. 4, we derive a tractable reward function:

$$r_t = \left(\sum_{i=1}^{N} \mathcal{H}(P_t(V_i))\right) - \left(\sum_{i=1}^{N} \mathcal{H}(P_{t+1}(V_i))\right) = \sum_{i=1}^{N} \left(\mathcal{H}(P_t(V_i)) - \mathcal{H}(P_{t+1}(V_i))\right). \quad (6)$$

Under our hard-constraint update model, the posterior entropy $\mathcal{H}(P_{t+1}(V_i))$ becomes zero for any newly constrained attribute $V_i \in \mathcal{V}_{\mathcal{E}_t}$, and remains unchanged for all other attributes. Therefore, the sum in Eq. 6 simplifies to include only the terms for the resolved attributes:

$$r_t = \sum_{V_i \in \mathcal{V}_{\mathcal{E}_t}} \mathcal{H}(P_t(V_i)). \quad (7)$$

This final equation states that the utility of an answer is the sum of the prior entropies of the attributes it clarifies. This signal is intrinsic, computationally efficient, and provides a robust foundation for optimizing the agent's inquiry policy. It is worth noting that we employ an unweighted sum of entropy reduction. We avoid manual weighting because Shannon entropy naturally embeds an "implicit statistical weighting": attributes with higher variance in the empirical prior yield larger information gain, automatically guiding the agent to prioritize statistically significant uncertainties without subjective heuristics.

## 3.2 OFFLINE POLICY OPTIMIZATION

With a defined reward signal, we can now train the agent's inquiry policy. Our approach is a fully offline process, which enhances stability and computational efficiency. It consists of two main stages: first, we generate a large-scale, static dataset of preference-ranked inquiries through simulation; second, we use this dataset to train the policy via an offline reinforcement learning algorithm.

**Automated Preference Data Generation** Our training process relies on a large-scale preference dataset, $\mathcal{D}_{\text{pref}}$, which we generate through an automated simulation framework. This simulation requires two key components: a "empirical prior" to provide prior probabilities (as in Eq. 2) and a set of ground-truth tasks. We construct both from a high-quality corpus of scientific diagrams, curated through a multi-stage filtering pipeline detailed in Figure 1.

Within the simulation, an "Oracle" agent, holding a ground-truth specification from our curated set, provides answers to inquiries proposed by multiple candidate models. The information gain for each inquiry is calculated via Eq. 7, yielding a training sample $\{p, \{r_1, \ldots, r_k\}, \{R_1, \ldots, R_k\}\}$, consisting of a prompt, a group of candidate responses, and their corresponding reward scores.

**Offline Adaptation of Group Relative Policy Optimization.** To optimize our policy $\pi_\theta$ on the static dataset $\mathcal{D}_{\text{pref}}$, we adapt the objective function from Group Relative Policy Optimization (GRPO) for an offline setting. While GRPO was originally proposed as an online algorithm that iteratively samples from the policy, we find its core objective is well-suited for offline training in our context. The rationale for this offline adaptation is twofold. First, the task of "asking a good question" is a capability already inherent in pretrained LLMs. The distribution of our generated candidate responses is therefore not expected to be drastically different from what the policy would generate, making on-policy sampling less critical. Second, using a static dataset eliminates the computational overhead of online generation, leading to a much more efficient and stable training pipeline.

For each group of responses, we first normalize the rewards into advantage estimates $A(r_i, p)$ via z-scoring within the group. This stabilizes the learning process across different prompts. Our offline algorithm then maximizes the following PPO-style clipped surrogate objective:

$$L_{\text{Policy}}(\theta) = \mathbb{E}_{(p, r_i, A_i) \sim \mathcal{D}_{\text{pref}}} \left[\min\left(\rho_i(\theta) A_i, \text{clip}(\rho_i(\theta), 1 - \epsilon, 1 + \epsilon) A_i\right)\right] \quad (8)$$

where the probability ratio $\rho_i(\theta) = \pi_\theta(r_i|p)/\pi_{ref}(r_i|p)$ measures the policy change against a frozen reference policy $\pi_{ref}$. The clipping function $\text{clip}(\cdot)$ constrains this ratio to a trusted region, preventing overly aggressive and destabilizing policy updates.

To further regularize the policy and ensure it does not deviate excessively from the pre-trained base model, we incorporate a KL-divergence penalty, leading to the final loss function:

$$L_{\text{total}}(\theta) = L_{\text{Policy}}(\theta) - \beta D_{KL}(\pi_\theta(\cdot|p) \,||\, \pi_{ref}(\cdot|p)) \quad (9)$$

where $\beta$ is a hyperparameter controlling the strength of the KL penalty. The log-probabilities $\log \pi(r|p)$ are computed autoregressively. To ensure the policy is only trained on its generation, we apply a loss mask so that the gradients are backpropagated only through the tokens of the response $r$, not the prompt $p$.

## 3.3 CONTRASTING METHODS FOR ABLATION STUDY

To rigorously evaluate the effectiveness of the **offline GRPO (OfG)** paradigm, we will use several other key baselines to train Nous for comparison in the experiments.

**Supervised Fine-Tuning (SFT)**: A baseline model fine-tuned only on the highest-reward (prompt, response) pairs from our dataset. This helps isolate the contribution of preference-based optimization over simple imitation learning. **Direct Preference Optimization (DPO)**: To compare against a prominent pairwise preference learning method, we implement a DPO baseline. DPO optimizes the policy to directly increase the log-probability ratio of preferred to dispreferred responses, using only the best and worst responses from each group. **Online GRPO (OnG)**: To validate the efficiency and stability of the offline approach, we also train a model using an online GRPO pipeline. This involves an initial SFT warm-up, followed by an iterative process of sampling responses from the policy, calculating their rewards, and updating the policy. All training methods ultimately include an SFT to train their ability for final integrated description.

## 4 EXPERIMENTS

We conduct a comprehensive set of experiments to evaluate our proposed framework. Our evaluation is designed to answer four key research questions: (1) Does our information-theoretic approach lead to more efficient interactions compared to established baselines? (2) Does higher interaction efficiency translate to superior quality in the final generated artifact? (3) Is the information gain-based reward signal the critical component of our framework's success? (4) How robust is the learned inquiry policy to variations in user expertise?

## 4.1 EXPERIMENTAL SETUP

**Models Under Evaluation.** Our primary model, Nous, is built upon Qwen3-8B and trained with full-parameter fine-tuning. For evaluation, we consider two groups of baselines. Trained Baselines: three Nous variants trained with alternative methods (SFT, DPO, OnG; see Section 3.3). Prompt-Based Baselines: a proprietary model (GPT-5: GPT-few ,GPT-zero) and a large open-source model (Qwen3-235B: Qwen-few, Qwen-zero), each tested under zero-shot and few-shot prompting. All prompts are instantiated using the *Socratic prompting* paradigm (Chang, 2023), which encourages the model to ask clarifying questions before producing a figure. We include these as the most relevant horizontal comparison, since no other mature baselines exist for scientific figure generation. Full prompt templates and hyperparameters are given in the Appendix.

**Evaluation Task and Data.** We take the task of scientific diagram generation in human-AI collaboration as our test scenario. The test data comes from a hold-out set of 100 complex real-world diagrams (see Figure1, for detailed sources see Appendix E). For each diagram, we simulate an interaction where the agent must elicit the complete specification from an Oracle. The Oracle, which holds the ground-truth specification for a target diagram and is configured identically to the one used for generating our training data. Each dialogue begins with a generic initial request, "I want to create a scientific diagram," and concludes when the agent indicates it has gathered sufficient information by outputting a final, consolidated description of the diagram. This automated simulation ensures a fair, controlled, and reproducible comparison across all models.

**Evaluation Metrics.** We employ a multifaceted evaluation strategy to assess both the process and the outcome: **Interaction Efficiency**: (1)We measure this by the average number of turns an agent takes to complete the dialogue, (2)and the cumulative information gain achieved throughout the interaction. Higher efficiency is indicated by fewer turns and a steeper information gain curve. **Output Quality**: We assess the quality of the final specification from two complementary angles: (1) subjective preference scores, where the final generated diagrams are evaluated by human and AI

Table 1: Experimental results of interaction efficiency, training resource consumption, and dynamic information gain.

| Model | Turns (↓) | Total IG (↑) | Resource hours(↓) | Information Gain Dynamics at Turn (↑) | | | | |
|---|---|---|---|---|---|---|---|---|
| | | | | Turn 1 | Turn 5 | Turn 10 | Turn 15 | Turn 20 |
| Nous (OfG) | 20.3 | **120.5** | 28.5 | 10.4 | 66.6 | **99.1** | **113.7** | **120.5** |
| Nous (OnG) | 22.0 | 115.8 | 93 | 7.8 | 59.4 | 88.4 | 107.2 | 114.3 |
| Nous (DPO) | 21.5 | 111.3 | 15 | **13.9** | 65.8 | 90.7 | 101.5 | 110.9 |
| Nous (SFT) | **17.1** | 94.3 | **13** | 12.6 | **78.1** | 90.5 | 94 | 94.3 |
| GPT-few | 22.6 | 93.5 | N/A | 9.1 | 60.4 | 77.4 | 88.1 | 92.1 |
| GPT-zero | 26.5 | 84.8 | N/A | 11.3 | 43.2 | 59.7 | 72.7 | 78.3 |
| Qwen-few | 19.5 | 90.5 | N/A | 10.6 | 61.1 | 76.4 | 85.9 | 90.5 |
| Qwen-zero | 25.3 | 81.5 | N/A | 6.6 | 48.1 | 64.4 | 77.2 | 80.3 |

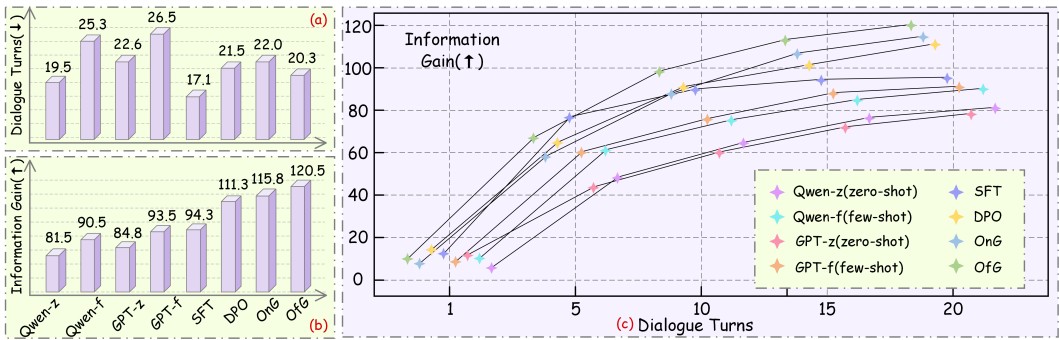

Figure 2: Experimental results of Interaction Efficiency. (a) The average number of dialogue turns for each model to complete information collection; (b) The average information gain obtained during the dialogue for each model; (c) The dynamic change of information gain during the dialogue

judges through pairwise comparisons, and (2) a suite of objective, specification-based metrics that quantitatively score the generated diagrams against the ground truth.

## 4.2 MAIN RESULTS

### 4.2.1 INTERACTION EFFICIENCY

**Dialogue Turns and Resource Cost**: Table 1 details the average number of dialogue turns and the associated training costs. First, all trained models complete the task in fewer turns than their non-trained counterparts. Then the performance of the SFT-trained agent shows the highest dialogue efficiency and the lowest training resources, but this brevity corresponds to the lowest total information gain among all trained models, indicating a premature and incomplete inquiry process. In contrast, the agent trained with OfG maintains a competitive turn count while requiring resources only marginally higher than DPO and significantly lower than OnG. This result highlights the scalability and cost-effectiveness of our offline training framework.

**Information Gain (IG) Dynamics**: Figure 2 plots the cumulative information gain against the number of dialogue turns, offering a more granular view of the inquiry strategies. The agents trained via OnG and OfG exhibit the most sustained information gain curves, demonstrating a robust ability to consistently pose high-value questions throughout the interaction. The SFT-trained agent, however, reveals a critical weakness: despite a strong start, its performance mirrors that of the non-trained models after the initial turns. They all fall into an "information bottleneck," where the ability to ask meaningful, probing questions sharply diminishes, causing their gain curves to flatten. This empirically validates the "frustrating guessing game" that motivated our work and underscores the necessity of a structured, goal-oriented training paradigm to overcome this fundamental limitation.

Table 2: Model win rate results under different tie-handling protocols: (1) "Win": ties not counted; (2) "W/T(0.5)": ties contribute 0.5; (3) "W/T": ties count as 1. All win-rate proportions are based on 400 pairwise judgments per model pair (100 prompts × 2 judges × 2 renderers); the standard error of a proportion is at most 0.025, so all 95% confidence intervals are within ±0.05.

| Model | 4o-image-1 | | | | | | nano-banana | | | | | |
| | Human Judge(↑) | | | GPT-5 Judge(↑) | | | Human Judge(↑) | | | GPT-5 Judge(↑) | | |
| | Win | W/T(0.5) | W/T | Win | W/T(0.5) | W/T | Win | W/T(0.5) | W/T | Win | W/T(0.5) | W/T |
|---|---|---|---|---|---|---|---|---|---|---|---|---|
| Nous (OfG) | **0.68** | 0.71 | 0.73 | **0.69** | **0.72** | **0.76** | **0.61** | **0.66** | **0.72** | **0.55** | 0.61 | 0.66 |
| Nous (OnG) | 0.69 | **0.72** | **0.75** | 0.63 | 0.67 | 0.71 | 0.60 | 0.65 | 0.70 | 0.54 | **0.61** | **0.67** |
| Nous (DPO) | 0.59 | 0.61 | 0.64 | 0.56 | 0.61 | 0.67 | 0.57 | 0.64 | 0.71 | 0.54 | 0.59 | 0.65 |
| Nous (SFT) | 0.49 | 0.51 | 0.53 | 0.42 | 0.48 | 0.55 | 0.38 | 0.48 | 0.57 | 0.37 | 0.48 | 0.59 |
| GPT-few | 0.45 | 0.47 | 0.50 | 0.45 | 0.52 | 0.58 | 0.34 | 0.44 | 0.54 | 0.39 | 0.47 | 0.56 |
| GPT-zero | 0.29 | 0.32 | 0.35 | 0.27 | 0.32 | 0.37 | 0.29 | 0.37 | 0.46 | 0.36 | 0.47 | 0.57 |
| Qwen-few | 0.36 | 0.40 | 0.43 | 0.35 | 0.42 | 0.48 | 0.35 | 0.42 | 0.49 | 0.32 | 0.41 | 0.49 |
| Qwen-zero | 0.23 | 0.27 | 0.30 | 0.24 | 0.28 | 0.32 | 0.26 | 0.33 | 0.40 | 0.28 | 0.36 | 0.44 |

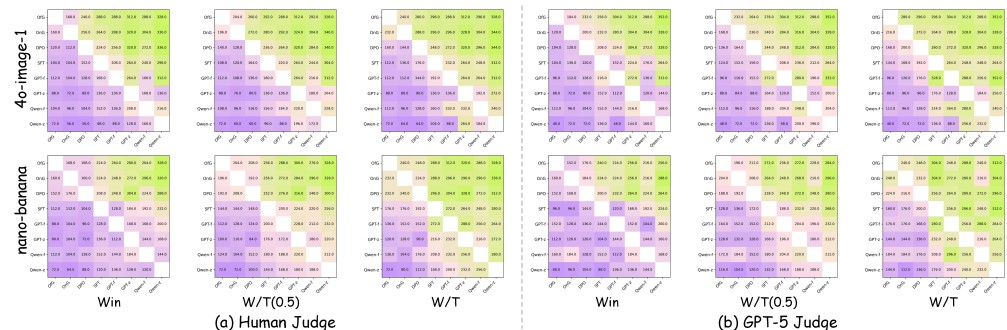

Figure 3: Model scores under different tie-handling protocols. (a) Results of human evaluation; (b) Results of GPT-5 model evaluation.

### 4.2.2 OUTPUT QUALITY

**Subjective comparison** We used two text-to-image backbone models (4o-image-1 and nano-banana) to generate two images based on the final natural language specifications of each model. These images were evaluated twice for their drawing quality through pairwise comparisons by human reviewers and GPT-5 with reference to the test set, resulting in a total of 11,200 comparisons. Table 2 reports the results under the evaluation protocols of three tie-handling methods, the results from the two judging protocols were highly consistent, lending reliability to the evaluation setup. Among the trained models, Nous trained with OfG and OnG achieved the highest win rates, outperforming DPO and SFT. The non-trained baselines lagged behind, with GPT-based models generally stronger than Qwen-based ones. Detailed pairwise results are visualized in Figure 3, and case studies are provided in the Appendix J.

**Objective metrics:** To complement these subjective judgments with reproducible quantitative scores, we employed the VisPainter framework, a tool that converts text specifications into editable vector graphics, with examples and descriptions provided in the Appendix E and J. This evaluates diagram specifications across six dimensions: Precision, Recall, Design Error Rate, Blank Ratio, Readability, and Alignment. Weighted score is calculated by applying weights of [0.2, 0.2, 0.2, 0.05, 0.25, 0.1] to these six dimensions, shown in Table 3 and Figure 4(a), highlight clear differences: OnG and OfG perform better in terms of drawing precision, recall, and readability. This is attributed to more detailed and information-rich image descriptions. The same applies to the blank ratio; thanks to more abundant component information, more efficient space utilization is achieved. Unexpected results were observed in terms of design error rate and alignment. This is because the

Table 3: Results of the final generated charts using the VisPainter framework. Higher scores in each item are better, and the design error rate has also been inverted to follow the same principle.

| Model | Precision | Recall | Design | Blank | Readability | Alignment | Score |
|---|---|---|---|---|---|---|---|
| Nous (OfG) | 0.83 | 0.84 | 0.51 | **0.83** | 0.79 | 0.88 | 0.76 |
| Nous (OnG) | **0.84** | **0.86** | 0.49 | 0.81 | **0.80** | 0.90 | **0.77** |
| Nous (DPO) | 0.80 | 0.81 | 0.52 | 0.78 | 0.75 | 0.87 | 0.74 |
| Nous (SFT) | 0.76 | 0.79 | 0.53 | 0.74 | 0.71 | 0.91 | 0.72 |
| GPT-few | 0.63 | 0.74 | 0.51 | 0.69 | 0.59 | 0.93 | 0.65 |
| GPT-zero | 0.42 | 0.77 | **0.55** | 0.61 | 0.41 | 0.93 | 0.57 |
| Qwen-few | 0.67 | 0.73 | 0.53 | 0.66 | 0.64 | 0.91 | 0.67 |
| Qwen-zero | 0.40 | 0.78 | 0.54 | 0.67 | 0.38 | **0.93** | 0.57 |

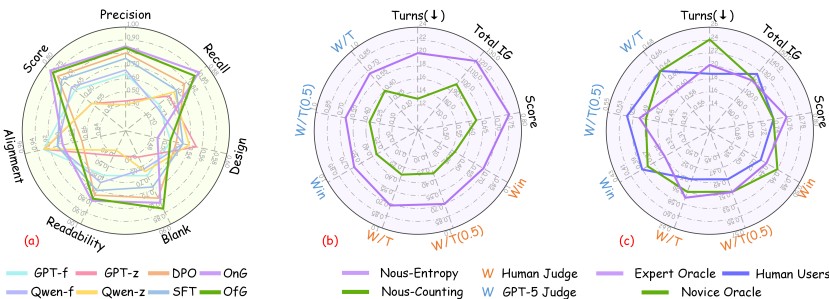

Figure 4: Visualization of experimental results. (a) Evaluation results of each model; (b) Results of ablation experiment 1; (c) Results of ablation experiment 2.

number of output elements is proportional to the chance of making mistakes during the drawing process, so SFT and prompt-based baseline models show higher scores in error rate and alignment. These patterns further confirm that models trained with principled inquiry signals have advantages over untrained models.

## 4.3 ABLATION STUDIES

**Reward Function:** To validate the critical role of our proposed information-theoretic reward signal, we conducted an ablation study. We replaced it with a heuristic-based "slot-counting" reward, which simply counts the number of specified attributes in each turn and treats all attributes equally. Using this new reward, we trained a model variant named Nous-Counting with the same OfG method on a dataset of identical scale, generated using the process from Section 3.2.

We evaluated this model under the identical experimental setup, with the results presented in Table 4 and 4(b). Nous-Counting completes dialogues in fewer turns, but it achieves substantially lower information gain and final output quality. This is because the slot-counting reward encourages a greedy policy that maximizes the quantity of resolved attributes, not their informational value. The model learns to ask broad, low-impact questions rather than strategically targeting high-entropy attributes first. This study confirms that our information-theoretic reward is essential for guiding the agent to learn an inquiry strategy that is not just superficially fast, but deeply effective.

**User Expertise:** Real-world collaboration involves users with diverse levels of expertise. We evaluated the robustness of Nous by testing it against three user personas: an Expert Oracle that uses precise technical terms (e.g., directed acyclic graph"), a Novice Oracle that uses vague, descriptive language (e.g., show it like a flowchart... with no loops"), and a group of ten participants representing real-world Human Users. To rigorously assess practical utility, we conducted human evaluations under two distinct settings: (1) *Zero-start*, where users describe their intents from scratch, and (2) *Draft-start*, where users provide an initial draft prompt containing partial information.

As shown in Table 4, Nous demonstrates strong adaptability across all user types. In the *Zero-start* setting, human users required fewer turns (17.3) than the Simulator (20.3) due to the higher information density of natural language, where users tend to disclose multiple attributes per turn. Most notably, the *Draft-start* setting demonstrated a significant efficiency leap, reducing the interaction to just 7.6 turns while maintaining high generation quality (Score 0.74). This highlights a central advantage of our Socratic framework: rather than relying on flawless user input, it strategically poses follow-up questions to progressively converge on the user's intent, proving effectively robust whether starting from vague descriptions or partial drafts.

Table 4: Quantitative evaluation of reward mechanisms and user adaptability. The table reports interaction turns, total Information Gain (IG), and win rates against human/GPT-5 judges. The upper section validates the superiority of our Information Entropy reward over a Counting baseline. The lower section demonstrates performance stability across different user levels, including simulated oracles and real humans starting from scratch (*Zero*) or partial drafts (*Draft*).

| Method | Turns ($\downarrow$) | Total IG ($\uparrow$) | Score ($\uparrow$) | Human Judge($\uparrow$) | | | GPT-5 Judge($\uparrow$) | | |
|---|---|---|---|---|---|---|---|---|---|
| | | | | Win | W/T(0.5) | W/T | Win | W/T(0.5) | W/T |
| Nous-Entropy | 20.3 | **120.53** | **0.76** | **0.68** | **0.70** | **0.72** | **0.60** | **0.63** | **0.66** |
| Nous-Counting | **13.6** | 97.11 | 0.63 | 0.28 | 0.30 | 0.32 | 0.34 | 0.37 | 0.40 |
| Expert Oracle | 20.3 | 120.53 | **0.76** | 0.41 | 0.50 | 0.59 | 0.33 | 0.49 | **0.65** |
| Novice Oracle | 24.1 | 122.47 | 0.74 | **0.42** | 0.49 | 0.57 | **0.38** | 0.51 | 0.64 |
| Human User_Zero | 17.3 | **126.44** | 0.75 | 0.40 | **0.51** | **0.61** | 0.38 | 0.49 | 0.61 |
| Human User_Draft | **7.6** | 42.31 | 0.74 | 0.42 | 0.50 | 0.59 | 0.37 | **0.51** | 0.64 |

## 5 CONCLUSION

This paper addresses a bottleneck in human-AI collaboration: "intention expression gap." We shift the paradigm from passive instruction-following to active, Socratic collaboration, introducing Nous, an agent that learns to resolve uncertainty about user intent through thoughtful inquiry. Our contribution is a training framework grounded in information theory, defining information gain as an intrinsic reward to eliminate costly human annotation and external reward models. We further show that Offline GRPO provides an efficient and stable path for training such agents. Experiments demonstrate that Nous achieves leading efficiency and output quality, while ablations confirm that the information-theoretic reward, rather than simple heuristics, is the decisive factor, and the agent remains robust across diverse levels of user expertise. In sum, this work presents a principled, scalable, and adaptive paradigm for resolving intent ambiguity, shifting the communication burden away from humans and moving us closer to AI partners capable of genuine collaborative thought.

**Reproducibility Statement** The models, prompts, data generation code, and model training code we used are all open-source. We have provided the code required to reproduce our research results in the supplementary materials. After the blind review period, we will release the complete code repository. To ensure the reproducibility of this paper, we have made efforts in the following aspects: (1) The code and data will be open-sourced once the paper is accepted. (2) We have conducted extensive experiments under different settings to verify the general applicability of the proposed framework. (3) We have provided a framework and evaluation methods based on open-source models, significantly improving reproducibility.

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

## 6 APPENDIX

## A STATEMENT ON LLMS USAGE

The authors used large language models (LLMs) during the writing process solely for language refinement and editing. It should be explicitly stated that LLMs were not employed in any core aspects of the research, including the formulation of research ideas, the design of methodologies, the execution of experiments, or the development of conclusions. All scholarly contributions were made independently by the authors.

## B EXTENDED DISCUSSION OF RELATED WORK

**Clarification and Inquiry as Strategy.** A growing body of work recognizes the strategic value of asking clarifying questions. In open-domain QA, for example, clarification has been shown to improve accuracy by resolving ambiguity before answering (Lee et al., 2023b). Other approaches model the decision of whether and when to ask a question based on the expected utility of future dialogue turns, effectively learning an optimal timing policy (Xu et al., 2024). In specialized domains like code generation, clarification also improves correctness, highlighting its broad value (Darji & Lutellier, 2025). While these methods validate the importance of proactive inquiry, they often optimize for single-answer correctness using heuristic signals or rely on downstream annotations to estimate future value. Nous shifts the focus from when to ask to what to ask. Our framework aims for convergence toward a complete, high-dimensional specification, where the reward is an immediate, intrinsic signal derived from entropy reduction over structured attributes, providing a stable, cumulative signal for optimizing the content of each inquiry.

**Information Gain as a Measure of Question Quality.** Our work builds on the principle of using information theory to quantify question value. In task-oriented dialogue, early frameworks used reward estimation to guide policy learning, though often as a proxy for external goals like booking success (Takanobu et al., 2019; Geishauser et al., 2021). More directly, work in visual dialogue has used information gain to explicitly model the value of "confirmation questions" (e.g., yes/no questions), demonstrating that such inquiries efficiently reduce the candidate set and improve success rates in guessing games like GuessWhat?! (Hu et al., 2024). Similarly, recent research establishes the "20 questions" game as a benchmark for active information seeking in LLMs, using expected information gain to rank and select the most discriminative question from a set of candidates generated via Chain-of-Thought prompting (Mazzaccara et al., 2024; Sahu, 2024). These studies collectively affirm that an entropy-based objective is a powerful tool for guiding efficient inquiry. However, directly applying existing information-theoretic methods to open-ended construction tasks faces significant challenges. For instance, UoT (Uncertainty of Thoughts) (Hu et al., 2024) relies on simulation-based planning, which becomes computationally intractable in our high-dimensional state space ($> 35,000$ combinations) due to the curse of dimensionality. Similarly, prompting-based clarification methods like CQ-Gen (Lee et al., 2023a) often focus on high-level semantic disambiguation rather than structural constraint resolution. In contrast, Nous integrates entropy reduction as a real-time intrinsic reward for a generative policy, avoiding the need for expensive full enumeration while maintaining precision in structural alignment. Nous integrates and advances these ideas into a scalable learning paradigm. Instead of using information gain as a post-hoc selection heuristic (Xiao et al., 2025) or applying it to a constrained set of question types, we use it as a real-time, intrinsic reward to train a generative policy. This enables Nous to learn to generate open-ended, natural language questions, which offers a significant advantage in high-dimensional, structured design spaces. In this way, we bridge the gap between the theoretical appeal of information gain and the practical challenge of training a proactive conversational agent for complex, creative tasks.

**Socratic Prompting versus Learnable Strategy.** Socratic prompting, exemplified by *Prompting Large Language Models with the Socratic Method* (Chang, 2023), encourages models to ask questions before answering through templates. *SocraticLM: Exploring Socratic Personalized Teaching* (Liu et al., 2024) extends this to personalized instruction, while *Hybrid Evaluation of Socratic Dialogue for Teaching* (Ilkou et al., 2025) evaluates its educational benefits and limits. While these approaches highlight the pedagogical value of Socratic interaction, they remain prompt-based or domain-specific. Nous extends the paradigm into a trainable policy: information gain defines the

objective, and offline preference data enables optimization. This transforms "asking questions" from prompt-driven behavior into a generalizable capability robust across user types.

**Comparison with Traditional Slot-Filling Paradigms.** While inspired by traditional Task-Oriented Dialogue (TOD), Nous addresses a fundamentally different problem scope. TOD systems typically perform extraction and retrieval: the user's intent is assumed to map to a specific entry in a database (e.g., a restaurant), and the system's goal is to fill static slots to filter this database. Nous, conversely, performs clarification and construction: the target (a scientific diagram) does not yet exist. The "slots" in our context are dynamic and interdependent (e.g., graph type dictates available attributes), and the goal is to align with a vague mental image rather than a database record. This distinction necessitates our shift from supervised slot-filling to reinforcement learning based on intrinsic information gain.

**Summary.** Prior work can be grouped into two broad directions: Socratic prompting methods that encourage proactive questioning through templates or pedagogy, and a method to achieve accurate question answering by quantifying the value of questions through entropy or mutual information. Nous advances both threads by combining structured belief states, closed-form entropy-based rewards, and offline policy optimization, thereby making clarification a scalable, principled, and generalizable strategy rather than a heuristic or template.

Table 5: Supplementary horizontal comparison experiments. As baseline models (UoT, CQ-Gen) cannot generate the finalized structural prompt for image generation, we compare only the interaction efficiency (Dialogue Turns) and the total Information Gain (Total IG). **Nous (OfG)** denotes our proposed method.

| Prompt Type | Method | Turns($\downarrow$) | Total IG($\uparrow$) |
|---|---|---|---|
| Promp_Zero | Nous(OfG) | 17.3 | 126.44 |
| | UoT | 33.6 | 38.38 |
| | CQ-Gen | 7.1 | 9.72 |
| Prompt_Draft | Nous(OfG) | 7.6 | 42.31 |
| | UoT | 37.6 | 12.30 |
| | CQ-Gen | 9.7 | 14.61 |

## C    SUPPLEMENTARY TO ABLATION EXPERIMENTS

To investigate the impact of training data quality on the final policy, we adjusted the data generation process. In addition to the Template Oracle, we introduced two variants: a Vague Oracle (providing incomplete information) and a Noisy Oracle (interjecting irrelevant information in its responses). Using these three data sources, we trained three respective models: Nous-Template, Nous-Vague, and Nous-Noisy. Distinct from the discussion on user expertise in Section 4.3, this section evaluates our framework's learning ability under different training data conditions.

The experimental results, shown in Table 6, reveal the following: **Adaptability to Vagueness.** Nous-Vague's performance in standard tests was comparable to the baseline model. This demonstrates the framework's effectiveness: although its training data (19,123 samples) was longer than the baseline data (11,851 samples) due to more clarification turns, leading to increased training time, the model still learned the core strategy of identifying high information-gain questions from these seemingly "inefficient" dialogues. **Filtering of Noise.** Nous-Noisy also performed nearly identically to the baseline model. This reveals a key property of our information-theoretic reward: it has a natural "immunity" to irrelevant information. Since noise cannot reduce the entropy of any attribute, its information gain reward is zero. Consequently, the training process automatically filters out the impact of noise, allowing the model to focus on learning genuinely effective question-answer patterns.

This study demonstrates our framework's high robustness to training data quality. Crucially, it also validates the robustness of our semantic parser, which successfully maps varied and imperfect responses back to the same underlying attributes, a key requirement for real-world application.

Table 6: Supplementary ablation study experimental results. Considering the significant time costs associated with data construction, model training, and drawing using VisPainter, the evaluation results of VisPainter are omitted in this experiment. Conduct model image generation evaluation experiments using nano-banana.

| Method | Turns ($\downarrow$) | Total IG ($\uparrow$) | Human Judge($\uparrow$) | | | GPT-5 Judge($\uparrow$) | | |
|---|---|---|---|---|---|---|---|---|
| | | | Win | W/T(0.5) | W/T | Win | W/T(0.5) | W/T |
| Nous-Template | 20.3 | **120.5** | 0.29 | **0.53** | **0.76** | **0.34** | **0.52** | 0.69 |
| Nous-Vague | 22.1 | 117.1 | **0.30** | 0.50 | 0.69 | 0.29 | 0.50 | **0.70** |
| Nous-Noisy | **19.7** | 115.8 | 0.26 | 0.48 | 0.70 | 0.33 | 0.49 | 0.65 |

# D  GENERALIZATION VALIDATION

**Experimental Setting.**    To test whether our framework generalizes beyond scientific diagram generation, we evaluate it in collaborative novel writing. This domain differs substantially from diagram creation in both task structure and interaction dynamics, yet retains properties that make systematic study feasible. Novel writing is open-ended and creative, but it is also composed of recurring elements such as characters, settings, and events. These elements can be represented as structured state vectors, enabling the construction of a empirical prior and the computation of per-turn information gain. At the same time, evaluation is relatively tractable: the quality of co-created narratives can be assessed through outline coverage and comparative judgments of readability and fidelity. These characteristics make collaborative novel writing another ideal testbed for examining the generality of our Socratic inquiry framework.

**Data Preparation and Training.**    We collect novels from publicly available corpora. Since long-form narratives are often lengthy and would substantially increase the workload, we simplify the data by selecting representative chapters as test material, which are further rewritten through AI-assisted editing to avoid copyright concerns. In total, we obtain 120 processed samples, with 100 used for training and 20 for testing. From each sample, we extract structured elements such as characters, settings, conflicts, and resolutions to form state vectors and construct a empirical prior as the prior. The data construction process follows the main text: the ground truth outline is provided to an Oracle, which answers model queries during simulation. Each question–answer pair is scored by information gain to create a preference dataset. Nous (OfG) is trained with offline GRPO, Nous (SFT) with supervised fine-tuning, and GPT baselines (zero-shot and few-shot: GPT-zero, GPT-fews) are included for evaluation.

**Evaluation Metrics.**    For evaluation, we adopt two dimensions consistent with the main paper: interaction efficiency and output quality. Interaction efficiency is measured by dialogue turns and total information gain, reflecting whether a model can ask high quality questions within a limited number of turns. Output quality is assessed through outline coverage and subjective quality evaluation. Specifically, we compare the generated summaries of novel passages using both human and GPT judges in pairwise evaluations. These metrics provide a balanced view of how effectively the models gather information and how well they translate it into coherent creative output.

**Results and Discussion.**    Novel writing represents a common and relatively structured domain, where LLMs already possess strong intrinsic capabilities. As shown in Table 7, this leads to notable efficiency for untrained models, which complete dialogues in fewer turns. However, Nous (OfG) achieves about 15% higher cumulative information gain compared to untrained baselines, confirming the benefit of entropy-based training. In terms of outline coverage, both OfG and SFT perform strongly, while GPT-few and GPT-zero show little distinction. For subjective evaluations by humans and GPT-5 judges, trained models consistently outperform baselines, though the margin is smaller than in our main domain. This may be due to the limited dataset size or the strong prior ability of LLMs in storytelling. Overall, the results validate that our framework retains effectiveness in a distinct creative domain, reinforcing its generalization capability and highlighting directions for future work in broader applications.

**Discussion: Applicability Boundaries and Marginal Utility.** While the experiments confirm the mechanism's universality, we observe a difference in the magnitude of improvement between the scientific diagram task (main paper) and the novel writing task. We attribute this difference primarily to the **High Baseline Effect**. Modern LLMs have internalized massive amounts of narrative structures during pre-training, providing them with a strong prior for storytelling. Even without active inquiry, baselines like GPT-4 can generate coherent narratives, leading to diminishing marginal returns for additional clarification. In contrast, scientific diagramming is an atypical generation task requiring precise spatial logic and strict constraints-areas where LLM priors are weak. Consequently, Nous delivers a qualitative leap in the diagram domain, whereas in the novel domain, it provides incremental optimization.

Based on these findings, we further define the **Applicability Boundaries** of our framework. We posit that the optimal operating zone for Nous is characterized by two key features: first, **High Structural Constraints**, where tasks possess objective logic (e.g., topological structures vs. pure brainstorming) that allows for accurate entropy calculation and efficient inquiry; and second, a **Significant Intention Gap**, where the user holds a specific, complex goal but struggles to articulate it. If a task allows for arbitrary open-ended generation where the user's intent itself is divergent, the value of eliminating uncertainty naturally decreases.

Table 7: Novel writing generalization experiment results. Dialogue efficiency and output quality are reported. All win-rate proportions are based on 80 pairwise judgments per model pair (20 prompts × 2 judges × 2 renderers).

| Method | Turns ($\downarrow$) | Total IG ($\uparrow$) | Coverage ($\uparrow$) | Human Judge($\uparrow$) | | | GPT-5 Judge($\uparrow$) | | |
|---|---|---|---|---|---|---|---|---|---|
| | | | | Win | W/T(0.5) | W/T | Win | W/T(0.5) | W/T |
| Nous (OfG) | 14.2 | **65.4** | **0.77** | **0.51** | **0.54** | **0.57** | **0.51** | **0.56** | **0.61** |
| Nous (SFT) | 11.1 | 60.7 | 0.73 | 0.49 | 0.51 | 0.53 | 0.44 | 0.51 | 0.57 |
| GPT-few | **10.4** | 57.8 | 0.68 | 0.43 | 0.46 | 0.50 | 0.40 | 0.48 | 0.56 |
| GPT-zero | 13.7 | 55.2 | 0.67 | 0.46 | 0.49 | 0.51 | 0.37 | 0.45 | 0.53 |

# E  DETAILED INTRODUCTION TO THE VISPAINTER FRAMEWORK AND IN-DEPTH ANALYSIS OF EXPERIMENTAL RESULTS

We incorporate the VisPainter framework to establish a quantitative evaluation pipeline, complementing the evaluations presented in Section 4.2.2. While subjective assessments by humans or AI focus on perceptual alignment, they lack granular quantification. VisPainter addresses this by providing an end-to-end process—from generation to execution—where output quality can be measured against specific, quantifiable indicators provided by VisBench. This integration serves as a critical supplement to our experiments, providing objective metrics to corroborate the effectiveness of our proposed method.

**Introduction to the VisPainter Framework** We adopt VisPainter as a baseline because it addresses a key limitation of diffusion-based text-to-image models: instead of producing rasterized bitmaps, it generates fully editable vector diagrams. This property is crucial for scientific illustration, where accuracy, semantic clarity, and iterative refinement are essential.

VisPainter is a multi-agent framework built on the Model Context Protocol (MCP), organized into three collaborative roles. The *Manager* parses intent and coordinates tasks; the *Designer* drafts and refines layouts; and the *Toolbox* provides over thirty MCP-wrapped atomic drawing operations. In our experimental setup, GPT-4o serves as the Manager and Gemini-1.5-Pro as the Designer, while the Toolbox handles structured execution. These roles collaborate to translate natural language instructions into structured, editable diagrams through iterative refinement.

Furthermore, the evaluation module within VisPainter is VisBench, a benchmark designed for scientific schematics. It provides seven evaluation metrics across four dimensions: accuracy, recall, design error rate, blank space rate, readability, alignment, and design steps. In our evaluation, we focus strictly on output quality, excluding the "design steps" metric. The VisBench dataset contains 360 entries, split evenly between T2I (Text-to-Image) and TI2I (Text-Image-to-Image) scenarios.

Our 100 test sets are selected from the T2I subset. This integration transforms VisPainter from a generative system into a rigorous research platform, ensuring reproducible and fair benchmarking. To the best of our knowledge, VisPainter was developed concurrently with our work, and its open-source release is forthcoming.

**In-depth Analysis of Metric Validity** In the results of Experiment 4.2.2, we observe a divergence in metric trends: while accuracy and recall improve significantly with our method, scores for design error rate and alignment show a slight decline. This phenomenon can be attributed to the intrinsic trade-off between information richness and execution complexity. Metrics such as *Recall* and *Accuracy* are positively correlated with information richness; as Nous captures more detailed constraints, the prompt becomes denser, naturally driving these scores higher. Conversely, metrics like *Alignment* and *Design Error Rate* are negatively correlated with task complexity. Since the capability of the backend designer (i.e., the plotting model) is fixed, increasing the number of components and structural constraints exponentially raises the execution difficulty. Untrained models often output simplistic diagrams with fewer elements, leaving little room for execution errors, which paradoxically results in higher "stability" scores. Therefore, the slight drop in these specific metrics reflects the increased challenge of rendering high-fidelity diagrams rather than a failure of the inquiry agent. Ultimately, the significant gains in semantic accuracy outweigh these minor execution artifacts.

## F    LIMITATIONS AND FUTURE WORK

**Limitations    The Attribute Independence Assumption:** For computational tractability, we assume conditional independence between attributes. Although this is a reasonable and effective first-order approximation, many real-world tasks involve complex dependencies; for instance, a specific layout choice might constrain the types of available components. We acknowledge that ignoring these correlations may lead to an overestimation of entropy, causing the agent to adopt a more conservative strategy (e.g., asking redundant confirmatory questions). However, this reduction to linear complexity is a necessary trade-off for real-time inference, avoiding the exponential overhead of modeling full coupling. Furthermore, our hard-constraint update mechanism ensures robustness by forcing the posterior probability to collapse upon explicit user feedback, thereby restricting the cost of this assumption to minor efficiency losses rather than systemic intention misalignment. Our current model does not explicitly model these interactions, leaving this as a promising direction for future work.

**Future Work**    The framework presented here has the potential to generalize to other structured domains, such as UI design, data visualization, or game creation. Beyond this broad applicability, two research directions are especially promising.

**Learning the Task Space:** Future agents could move beyond a fixed attribute set by inferring relevant attributes and their structure directly from interaction or large dialogue corpora. This would allow the framework to adapt dynamically to new tasks without manual specification.

**Toward Mixed-Initiative Dialogue:** Our current model is agent-led. A natural extension is to support mixed-initiative collaboration, where users proactively contribute information and the agent must decide whether to integrate it or pivot its strategy. This would yield more natural and adaptive interaction.

Together, these directions point toward making inquiry-driven collaboration more generalizable and human-like.

## G    IMPLEMENTATION DETAILS

**Training Environment and Hyperparameters**    All models were trained using a full-parameter fine-tuning approach on a high-performance computing cluster equiped with 8x NVIDIA H200 (141GB) GPUs. We utilized bfloat16 mixed-precision training to optimize for speed and memory efficiency. The key hyperparameters used for training each of the models are detailed in Table 8. We selected these parameters based on preliminary experiments to ensure stable and effective training for each respective method.

Table 8: Hyperparameters for SFT, DPO, OnG, and OfG.

| Hyperparameter | SFT | DPO | OnG | OfG |
|---|---|---|---|---|
| *Model & Data Configuration* | | | | |
| Base Model | | Qwen3-8B | | |
| Fine-tuning Method | | Full-parameter | | |
| Training Precision | | bfloat16 | | |
| Max Sequence Length | | 4096 | | |
| *Optimization* | | | | |
| Optimizer | | AdamW | | |
| Learning Rate (lr) | 1e-6 | 1e-6 | 1e-6 | **1e-6** |
| LR Scheduler Type | | Cosine | | |
| Warmup Steps | 50 | 50 | 50 | **50** |
| Epochs | 5 | 5 | 5 | **5** |
| Batch Size (per device) | 1 | 1 | 1 | **1** |
| Gradient Accum. Steps | 2 | 2 | 2 | **2** |
| Weight Decay | 0.01 | 0.01 | 0.01 | **0.01** |
| *Regularization & RL-specific* | | | | |
| KL Coefficient ($\beta$) | N/A | 0.1 | 0.01 | **0.01** |
| PPO Clip Epsilon ($\epsilon$) | N/A | N/A | 0.2 | **0.2** |

## H DATASET DETAILS

Our dataset was constructed from a corpus of scientific papers sourced from arXiv and PubMed Central (PMC), covering a wide range of disciplines to ensure diversity. The primary arXiv categories included Computer Science (43.1%), Physics (22.7%), Quantitative Biology (14.8%), Electrical Engineering (11.5%), and others such as Economics and Statistics (7.9%). All source materials were confirmed to be under open-access licenses (e.g., Creative Commons, arXiv.org non-exclusive license) that permit reuse for research. The initial pool of approximately 1 million figures was refined through a multi-stage pipeline: an initial filtering with CLIP to remove data plots, followed by a fine-grained selection of schematic diagrams using Qwen-2.5-VL-72B. A final manual verification by three domain experts ensured the relevance and quality of each diagram, resulting in a curated set of 1,100 figures. Of these, 1,000 were used for training and 100 were held out for testing. To ensure the reproducibility of our experiments involving proprietary models, all API calls for data generation and evaluation were made using model versions available after 4-14-2025.

## I PROMPTS

All prompt templates, data construction, model training, and result evaluation codes are included in the attachments submitted along with the article. Here we provide the Socratic prompting templates used for our zero-shot and few-shot baselines. The core idea is to encourage the assistant to proactively ask clarifying questions before finalizing the diagram specification:

### I.1 ZERO-SHOT SOCRATIC PROMPTING

```
"You are an assistant that helps design scientific diagrams.
Do not produce the diagram immediately. Instead, follow these steps:
1. Ask the user a clarifying question about the diagram (e.g., type,
    layout, number of components, connections, or style).
2. Continue asking such clarifying questions until enough information has
    been gathered to produce a complete diagram specification.
3. Only after clarification is complete, summarize the final diagram
    specification in a structured format (JSON).
Remember:
- Ask focused, concrete questions (one per turn).
- Avoid vague or open-ended questions.
- The final specification must be complete and self-contained."
```

## I.2 FEW-SHOT SOCRATIC PROMPTING

```
"You are an assistant that helps design scientific diagrams.
Do not produce the diagram immediately. Instead, follow these steps:
1. Ask the user a clarifying question about the diagram (e.g., type,
    layout, number of components, connections, or style).
2. Continue asking such clarifying questions until enough information has
    been gathered to produce a complete diagram specification.
3. Only after clarification is complete, summarize the final diagram
    specification in a structured format (JSON).
Remember:
- Ask focused, concrete questions (one per turn).
- Avoid vague or open-ended questions.
- The final specification must be complete and self-contained.
Here are two example questions for your reference:Example 1: Example 2:"
```

## J   DATA AND ILLUSTRATION EXAMPLE

**Data Example   Example of offline GRPO training data based on information entropy:**

```
"   {
     "prompt": "User: I want to create a scientific diagram.\nAssistant:
    What specific element or component should serve as the target
    identifier (connection.to_id) for the connections ...",
     "responses": [
       "What is the primary scientific concept this diagram aims to
    illustrate?",
       "What is the primary scientific concept or model you intend to
    illustrate with these components and connections?",
       "What is the fundamental scientific concept or process that your
    diagram is intended to illustrate?",
       "What primary scientific concept or system does this diagram aim to
     illustrate?",
       "What is the primary scientific concept or system that this diagram
     is intended to illustrate?",
       "What overall visual structure do you envision for your diagram?",
       "Which overall diagram type do you envision such as a flowchart
    ...?",
       "What specific element or component should ...?"
     ],
     "reward": [
       16.83,
       13.91,
       10.56,
       8.21,
       6.78,
       4.09,
       1.32,
       0.0
     ]
   },"
```

**Example of offline GRPO training data based on slot-counting:**

```
"     {
     "prompt": "User: I want to create a scientific diagram.",
     "responses": [
       "What is the primary scientific concept this diagram aims to
    illustrate?",
       "What is the primary scientific concept or topic that this diagram
    is intended to illustrate?",
       "What is the primary scientific concept or topic that this diagram
    aims to illustrate?",
```

```
     "What are the main components of your scientific diagram, and what
   unique identifiers (component.id) will you assign to each?",
     "What specific components (component.id) do you envision in your
   diagram, and what identifiers or labels should be assigned to each to
    clarify their roles?",
     "Which key components (nodes) do you envision for your diagram, and
    how would you uniquely identify each (i.e., what are their
   respective component IDs)?",
     "Can you identify the distinct components for your diagram by
   assigning specific IDs or names, and briefly describe the role of
   each?",
     "What are the main components (component.id) you envision including
    in your scientific diagram, and what specific role does each play in
    illustrating the concept?"
   ],
   "reward": [
     4.0,
     2.0,
     2.0,
     1.0,
     1.0,
     1.0,
     1.0,
     1.0
   ]
 }," 
```

**Objective metrics**    This section presents drawing examples generated using the VisPainter framework, as shown in Figure 5.

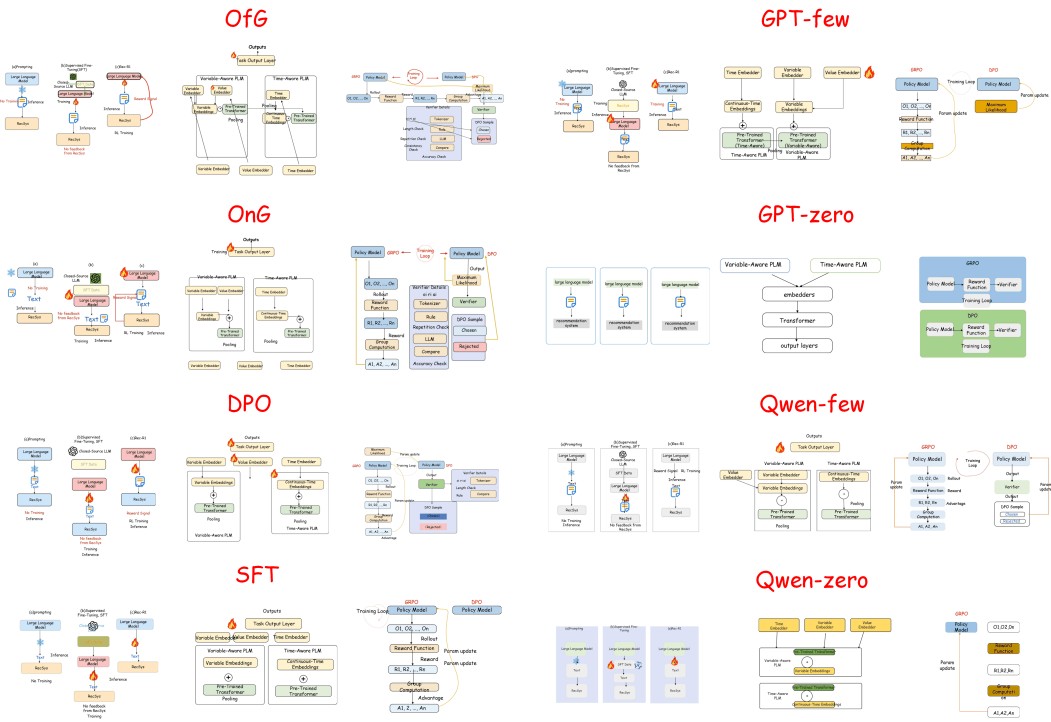

Figure 5: This section presents drawing examples generated using the VisPainter framework

**Subjective comparison** This section presents drawing examples generated by two models (4o-image-1 and nano-banana), as shown in Figure 6.

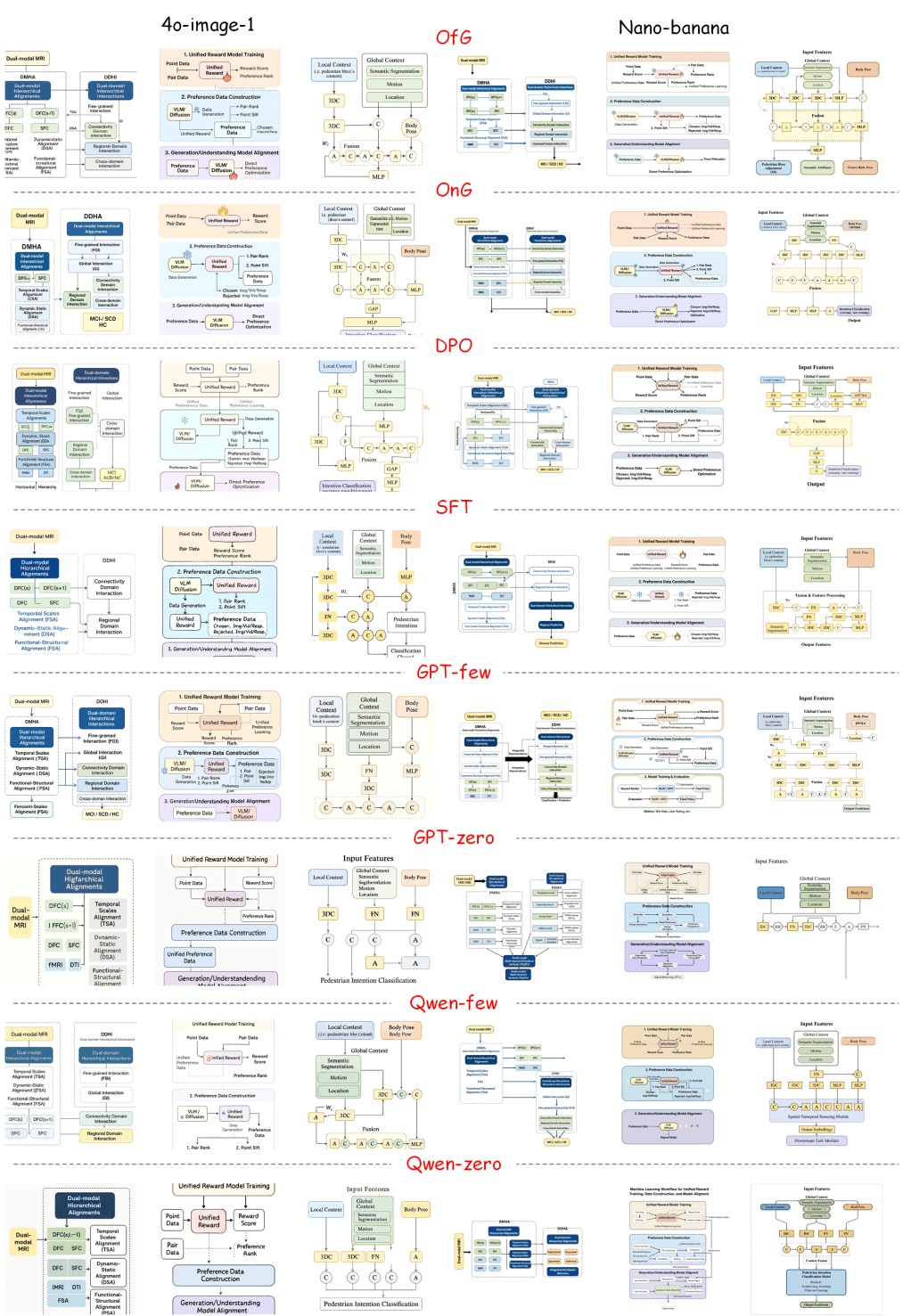

Figure 6: Partial Examples of Model-Generated Images

