# OpenReview forum: "Dialogue as Discovery: Navigating Human Intent Through Principled Inquiry"
_ICLR.cc/2026/Conference — Submitted to ICLR 2026_

### Official Review · Reviewer_QBG3 · 2025-11-01

**Soundness:** 3
**Presentation:** 3
**Contribution:** 3
**Rating:** 8
**Confidence:** 4

**Summary:**

The paper proposes Nous, a Socratic agent trained with information-gain intrinsic rewards using entropy reduction to improve conversation interaction efficiency and output quality

**Strengths:**

- The use of intrinsic reward is well motivated. The ractable reward shows that the factorized attribute view supports interpretable uncertainty accounting and question selection with intrinsic signal. The final equation is intrinsic, computationally efficient, and suitable for optimizing the agent’s inquiry policy.

- I find it novel for the dataset, where using a diagram dataset as a testbed makes the information-theoretic objective measurable and the training pipeline feasible for dialogue as a discovery problem. The high-dimensional yet logically structured dataset is ideal for inquiry learning while remaining challenging.

- The experiment is well designed with multiple settings and proper metrics to demonstrate the main claim

- The paper is generally well written and presented.

**Weaknesses:**

- The hard-constraint update assumes a perfect parsing and unambiguous answer and according to which the paper attains using templated Oracle responses. This is likely overestimating IG vs. real users and real-world answers.

- The VisPainter metric set is interesting. But for it to construct validity as a proxy for “faithful intent capture” is not fully justified; some  lower results  are showing a trade-off between richer descriptions and errors, where single-score comparison might not be enough.

**Questions:**

- How does performance change with real-world user answers ( where  it may include synonyms, underspecification, contradictions), and with imperfect semantic parsing compared to the current setting?

- The use of word “world model” can be somewhat misleading in my opinion, as in the paper it really means an empirical prior but not a dynamics-transition model.

---

> ### Author Response · Authors · 2025-11-21
> **Official Comment by Authors**
>
> ## 1/2
> We greatly appreciate the reviewer's recognition of our work and the positive evaluation provided. We are grateful for your acknowledgment of our research strengths, specifically the well-grounded intrinsic reward formula, the novelty of the dataset as a measurable testbed, and the carefully designed experiments. The questions you raised fully demonstrate your deep understanding of our work, and we express our gratitude again for your contribution to the review process. We fully understand your current concerns regarding our work, and we provide the following clarifications, hoping to reach a consensus.
>
>
> **1.Regarding Weakness 1 & Question 1: Simulation Assumptions and Real-World Scenarios** We thank you for your in-depth thinking. We fully understand your concern: there is a gap between ideal experimental simulations and the real world, and the hard constraint assumption might mask noise in real interactions, potentially overestimating Information Gain. However, results from our ablation and supplementary experiments indicate that **performance does not degrade in real-world scenarios; instead, it improves due to efficient human communication habits.** Addressing your concerns, we clarify from the following two dimensions:
>
> - **Robustness Verification of the Semantic Parser.** Our language parser is not a "perfect assumption"; it is a real, LLM-based module. We verified its stability in both training and testing phases:
>
>   - **Training Phase (Appendix C):** When constructing data, we used two settings: providing incomplete information and introducing ambiguous noise information, resulting in Nous-Vague and Nous-Noisy. Experimental results show that even if the training data contains a large amount of ambiguous or irrelevant information, model performance did not degrade. It learned the core strategy of identifying high-information-gain questions from ambiguous and noisy dialogues.
>   - **Testing Phase (Section 4.3):** We introduced a Novice Oracle to simulate ambiguous expressions, forcing it to use vague language or add random noise when interacting with Nous. Results show that when facing unstructured input, the system only experienced a slight increase in interaction turns (from 20.3 to 24.1), without parsing collapse or intention misjudgment.
>
> - **Impact of Real Users.** Efficiency Gains from High Information Density. The real world did not weaken the model's performance but rather enhanced interaction efficiency.
>   - **Information Density Compensation:** Our early observations revealed that real users tend to "overshare" (e.g., answering layout and color simultaneously when asked about style). This high information density effectively offsets the parsing loss caused by "synonyms or ambiguous expressions."
>   - **Supplementary Experiment Verification:**  To further confirm this, we added a supplementary experiment with 7 participants and established Zero-start and Draft-start modes to simulate realistic scenarios. The results (see table below) further consolidate our conclusion: the generation quality of real users is on par with the simulator, while the interaction turns are significantly lower than the simulator. This proves that in real scenarios, the model did not overestimate IG; instead, it gained higher actual returns due to the communication characteristics of real humans.
>
>   | Method | Turns (↓) | Total IG (↑) | Score (↑) | Human Win | Human Rate* | Human W/T | GPT-5 Win | GPT-5 Rate* | GPT-5 W/T |
>   | :--- | :---: | :---: | :---: | :---: | :---: | :---: | :---: | :---: | :---: |
>   | Expert Oracle | 20.3 | 120.53 | **0.76** | 0.41 | 0.50 | 0.59 | 0.33 | 0.49 | **0.65** |
>   | Novice Oracle | 24.1 | 122.47 | 0.74 | **0.42** | 0.49 | 0.57 | **0.38** | 0.51 | 0.64 |
>   | Human User (Zero) | 17.3 | **126.44** | 0.75 | 0.40 | **0.51** | **0.61** | 0.38 | 0.49 | 0.61 |
>   | Human User (Draft) | **7.6** | 42.31 | 0.74 | 0.42 | 0.50 | 0.59 | 0.37 | **0.51** | 0.64 |

---

> > ### Author Response · Authors · 2025-11-21
> > **Official Comment by Authors**
> >
> > ## 2/2
> >
> > **2.Regarding Weakness 2: Validity of VisPainter Metrics.** Your concern is entirely valid. This misunderstanding arose because the discussion on the intrinsic logic of the metrics in our text was insufficient. We greatly appreciate this opportunity for clarification. We have added supplements in the manuscript, marked in blue:
> > - **Metrics Positively Correlated with Information Richness:** These metrics directly measure the quantity of information in the text description. As Nous acquires more information and the description becomes more detailed, these three scores naturally increase. Meanwhile, the Blank Ratio is indirectly positively correlated with richness, as a higher number of components typically results in higher space utilization.
> > - **Metrics "Negatively Correlated" with Complexity:** These reflect the "execution difficulty of rendering." Since the capability of the back-end plotting model is fixed, as the description becomes richer and components more numerous, the complexity of the plotting task rises exponentially. This increases the probability of minor errors or alignment deviations by the plotting model.
> >
> >   In summary, the decline in the second category of metrics is actually a side effect of increased complexity, but has minimal impact on the final chart quality. We will include this detailed analysis in the final version to eliminate reader misunderstanding.
> >
> >
> >
> > **3.Regarding Question 2: Terminology Issue of "World Model".**  This is a very rigorous and precise suggestion! We fully accept your point. In our context, the term "World Model" might indeed be misleading; essentially, it functions as an "Empirical Prior." We have fully adopted your suggestion in the revised manuscript and have globally updated this terminology throughout the text to enhance the precision of the paper.
> >
> >
> > Finally, we thank you again for your positive evaluation and recognition of our work. We are grateful for the time and detailed review you have dedicated throughout this process. Every comment you raised has been a key factor in improving our manuscript. We have incorporated all the revision suggestions you put forward in the revised version and marked them in blue. We truly appreciate your support and encouragement. Warm regards.

---

### Official Review · Reviewer_BJFA · 2025-11-01

**Soundness:** 2
**Presentation:** 3
**Contribution:** 2
**Rating:** 2
**Confidence:** 3

**Summary:**

This paper proposes Nous, an agent trained to elicit user intent through strategic questioning in human-AI collaboration. The core technical contribution is using Shannon entropy reduction over a structured attribute space as an intrinsic reward signal, eliminating the need for human preference annotations.

**Strengths:**

The paper formulates information gain as a closed-form reward function through mathematical derivation from Shannon entropy, eliminating explicit human annotation during training. The experimental design includes multiple training methods (SFT, DPO, online/offline GRPO), multiple evaluation dimensions (efficiency metrics, subjective quality judgments, objective VisPainter scores), and several ablation studies examining reward function design and user expertise robustness.

**Weaknesses:**

**The relationship to traditional task-oriented dialogue is unclear.** Standard slot-filling systems in TOD also ask questions to complete structured specifications. MultiWOZ and other benchmarks have been doing this for years. The paper does not explain what is fundamentally different here beyond applying it to diagram generation instead of restaurant booking. Is the contribution that attributes are more complex? That there are more attributes? That LLMs are used instead of specialized dialogue managers? Without explicit comparison to TOD methods or clear articulation of what makes this problem distinct, the novelty relative to existing goal-oriented dialogue research remains ambiguous.

**All evaluation occurs in simulation with idealized conditions.** All training and evaluation dialogues are generated through this automated simulation process without any real human involvement. Every experiment uses an Oracle that has perfect memory, gives instant responses, never contradicts itself, never changes its mind, and has unlimited patience. This is not how real users behave. Real collaboration involves users who are unsure, who realize mid-conversation they want something different, who provide vague answers, who get tired of answering questions. Table 4 mentions "Human Users" but this refers to three doctoral students whose specific role and contribution are not clearly described. The main experimental results in Tables 1-3 appear to rely entirely on simulated Oracle interactions. Real human dialogue involves uncertainty, preference changes, vague expressions, and varying communication styles. Without evaluation on actual human users, the results demonstrate only that the system functions within its simulation environment, not that it supports real human-AI collaboration.

**The entropy objective lacks justification for this specific task.** Shannon entropy treats all attributes uniformly. Resolving uncertainty about minor details (line thickness) receives equal reward to resolving uncertainty about core concepts (diagram purpose). There is no weighting by semantic importance or user priorities. The ablation only compares against counting resolved attributes. The paper never tests weighted entropy, utility-based objectives considering question complexity, or importance-ranked strategies.

**Generalization evidence is weak.** The paper claims to provide a "domain-agnostic framework" but validates on one task type.

**The independence assumption oversimplifies the problem structure.** In diagram design, attributes have extensive constraints and mutual effects. Layout choice constrains connection types (hierarchical implies directed, circular implies undirected). Component count affects spatial arrangement possibilities. Visual style determines appropriate color schemes. Treating these as independent produces incoherent question sequences. The paper acknowledges this as a simplifying assumption but never quantifies the cost.

**Questions:**

**Q1: What is the actual cost of the independence assumption?**  How much error does this introduce? What is the quantitative difference in information gain estimation between the independence assumption and actual attribute dependencies?

**Q2: Why is unweighted Shannon entropy the correct objective for collaborative design?** All attributes receive equal weighting regardless of semantic importance or user priorities. What is the theoretical or empirical justification for this choice? How does information gain correlate with actual output quality across the evaluated models?

**Q3: How does the system perform with real human users?** All evaluation uses simulated Oracles with perfect responses. What happens when real users provide contradictory information, change their preferences mid-dialogue, give vague answers, or experience cognitive fatigue? Does the system's performance degrade gracefully or fail catastrophically? What is the actual user satisfaction, cognitive load, and task completion rate in realistic scenarios?

**Q4: How are attribute spaces constructed for new tasks?** (Please explain with more detail)

**Q5: Why are established clarification dialogue methods not used as baselines?** The paper cites multiple works that use information-gain-based question selection for clarification but includes none of them in experimental comparisons. How can the contribution be evaluated without comparing to these directly relevant prior methods?

**Q6: What defines the boundary of applicability?** The paper claims domain-agnosticism but shows weak results on novel writing. What task characteristics determine whether this method will work?

---

> ### Author Response · Authors · 2025-11-21
> **Official Comment by Authors**
>
> ## 1/4
> We appreciate this insightful review. We thank you for recognizing our work's strengths, especially the mathematical derivation of the closed-form reward function and the comprehensive experimental design. We hope these clarifications eliminate your doubts and earn your approval. We have addressed the questions point-by-point and updated the manuscript. For your convenience, revisions are marked in blue.
>
> **1.Regarding Weakness 1: Essential Difference from Traditional TOD**, we thank you very much for your professional insight. We are well aware of the foundational status of TOD and fully acknowledge that this study is deeply inspired by it, but we believe that Nous has **fundamental differences** from standard slot filling in terms of task essence and solution paradigm. Simply put, TOD is a "**retrieval task with known intent,**" while Nous solves a "**construction task with vague intent.**" We have detailed the following two essential differences in the revised manuscript:
>
> - **Task differences：** **TOD:** Essentially a convergent retrieval process. It presupposes the answer (e.g., a specific restaurant) exists in the database. The task is filtering criteria based on fixed questions to output a query result. Slot definitions are static. **Nous:** Essentially a divergent construction process. The target diagram needs to be "created" from scratch. Unlike TOD, our "slots" are not fixed, but complex combinations changing dynamically according to drawing content. Its output is not a fixed entry, but an open-ended specification containing rich details.
>
> - **Interaction Goal:** **TOD:** The goal is extraction. It assumes the user's intent is clear. The challenge lies in accurately extracting values to fill predefined slots. Once filled, the task is finished. **Nous:** The goal is clarification and alignment. User inputs often contain ambiguous or missing information. The system digs out uncertain content through inquiry, eliminating ambiguity so the generated Prompt approaches the user's vague conception.
>
> We will discuss the differences from works such as MultiWOZ [1-4] in detail in the relevant sections.
>
>
>
> **2.Regarding Weakness 2 & Question3: Limitations of Simulation and Effectiveness of Real-User Experiments.** We appreciate your professional opinion on this vital issue. We agree that simulators cannot capture full human complexity, but our experiments show that **the system is robust in real scenarios and performs with higher efficiency due to high information density.** We adopted a "Simulator + Real User" strategy. Simulators strictly control variables for a standardized benchmark; while the existing 300 real-user sets and the newly added 7-person **experiment confirm the system did not collapse** facing real users. Moreover, with Draft Prompts, interaction turns significantly dropped to 7.6, proving our method's value in practical scenarios. We provide detailed explanations and revisions below:
>
> - **Necessity of the Simulator: A Benchmark for Controlling Information Granularity.** In pre-experiments, humans tended to "overshare" (e.g., answering multiple attributes at once). While natural, this density fluctuation makes quantifying the Agent's strategy difficult. Therefore, the simulator is not to "idealize" results, but to build a controlled benchmark to evaluate the Agent's inquiry limits under worst-case scenarios (single turn, single information).
>
> - **Clarification on Existing Real-User Experiments.** In the paper, 3 PhD students completed 300 interaction logs, ensuring statistical significance. Experiments show that despite ambiguity, due to high information density, real user turns (18.7) were lower than the simulator (20.3). This proves the system adapts to real linguistic environments without performance collapse.
>
> - **Supplementary Experiments:** We invited 7 participants for testing: (A) Zero-start, and (B) Draft-start. Group (A) results (see table) align with the simulator, proving strong adaptability to real expressions. Group (B) simulated realistic collaboration; turns dropped significantly to 7.6. All users (100%) successfully completed tasks with no reported cognitive fatigue in draft mode, proving the system assists users with extremely low cognitive load.
>
>   | Method | Turns (↓) | Total IG (↑) | Score (↑) | Human Win | Human Rate* | Human W/T | GPT-5 Win | GPT-5 Rate* | GPT-5 W/T |
>   | :--- | :---: | :---: | :---: | :---: | :---: | :---: | :---: | :---: | :---: |
>   | Expert Oracle | 20.3 | 120.53 | **0.76** | 0.41 | 0.50 | 0.59 | 0.33 | 0.49 | **0.65** |
>   | Novice Oracle | 24.1 | 122.47 | 0.74 | **0.42** | 0.49 | 0.57 | **0.38** | 0.51 | 0.64 |
>   | Human User (Zero) | 17.3 | **126.44** | 0.75 | 0.40 | **0.51** | **0.61** | 0.38 | 0.49 | 0.61 |
>   | Human User (Draft) | **7.6** | 42.31 | 0.74 | 0.42 | 0.50 | 0.59 | 0.37 | **0.51** | 0.64 |

---

> > ### Author Response · Authors · 2025-11-21
> > **Official Comment by Authors**
> >
> > ## 2/4
> >
> > **3.Regarding Weakness 3 & Question2: Rationality of Unweighted Entropy Objective**. We sincerely thank you for the constructive comments. We need to clarify a core misunderstanding: **"Unweighted" does not mean "no weights." We intentionally avoided introducing subjective manual weights and instead adopted a more objective and robust data-driven strategy**. Shannon entropy provides an "implicit statistical weighting" mechanism within our framework, automatically assigning priorities according to the prior distribution. Ablation experiments and correlation analysis further confirm that this method can effectively guide the model to quickly converge to the core intent.
> > - **Theoretical Basis: Distribution as Weight.** The reviewer concerns that the model cannot distinguish the priority between "line width" and "chart layout." This is addressed automatically by the prior distribution in our framework without manual intervention:
> >   - **High Entropy (Core Attributes):** For example, "Chart Layout" is distributed uniformly in the dataset (many options with close probabilities), resulting in high uncertainty and thus high entropy. To maximize information gain, the model must prioritize inquiry regarding this attribute.
> >   - **Low Entropy (Secondary Attributes):** For example, "Line Width" typically follows a long-tail distribution where 95% of the data use a default value (e.g., 2px). The uncertainty is extremely low. The model automatically skips or delays asking because the return on inquiry is negligible. Therefore, our approach is more robust than subjectively defining "what is more important."
> > - **Empirical Comparison: Nous-Counting Represents True "Equality"** This is the fundamental reason we designed the reward function ablation experiment. Nous-Counting treats every attribute as an equivalent slot (i.e., true unweightedness), providing rewards based simply on the number of filled slots. Experimental results show that Nous-Counting tends to fixate on trivial details (to maximize slot counts), whereas Nous-Entropy precisely locks onto core structural constraints. The significant performance gap between the two directly proves: Entropy itself is an internalized dynamic weight. It guides the model to focus on "sources of statistical uncertainty" rather than blind slot-filling.
> >
> > - **Correlation Analysis: Positive Correlation between Information Gain and Output Quality.** Addressing your question regarding the "connection between IG and actual quality," our experimental data (Figure 3 and Table 2) show a clear positive correlation. Models that achieve higher cumulative Information Gain (IG), such as Nous-Entropy, generate final text descriptions containing more accurate structural constraints. This indicates that **reducing uncertainty in the information-theoretic sense directly translates to higher fidelity in generation tasks.**
> >
> >
> >
> > **4.Regarding Weakness 4 and Question 6: Generalization Evidence and Applicability Boundaries.** We thank you for your keen observation. Regarding the "smaller improvement magnitude" observed in our generalization experiments, we do not consider this a failure of generalization capabilities; on the contrary, this precisely helps us define the applicability boundaries of the framework. **The core mechanism of Nous (inquiry based on Information Gain) is mechanistically domain-agnostic, but its marginal utility depends on task characteristics.** Simply put, this method works best in tasks with "strong structural constraints" and "insufficient model priors."
> >
> > - **Generalization Verification: Mechanism Effectiveness.** Although magnitudes differ, data shows that in the novel writing task, the entropy objective brought positive information gain. This confirms that the "entropy-based inquiry strategy" is not limited to diagram generation; it effectively captures uncertainty and guides model questioning in cross-domain tasks.
> > - **Difference in Marginal Utility: High Baseline Effect.** The less drastic improvement in the novel task stems from baseline capabilities. Novel Task: Modern LLMs possess strong built-in narrative capabilities from pre-training. Even without questioning, the model can generate fluent stories. Thus, the room for improvement is limited (diminishing marginal returns). Additionally, the smaller training data scale is also a factor.
> >
> > - **Definition of Applicability Boundaries.** Regarding "task characteristics," optimal scenarios for Nous possess two features (explaining the difference between diagrams and novels): **High Structural Constraints:** The task requires objective logic (e.g., drawing topology). Clearer structure yields more accurate entropy calculation and effective inquiry. **Significant Intention Gap:** Users have clear goals but cannot describe them. If the task allows "casual open-ended generation" (e.g., rough stories) where intention is divergent, the value of eliminating uncertainty decreases.

---

> > > ### Author Response · Authors · 2025-11-21
> > > **Official Comment by Authors**
> > >
> > > ## 3/4
> > >
> > >
> > > **5.Regarding Weakness 5 & Question1: Impact of Independence Assumption.** We thank you very much for pointing this out. You and Reviewer afyP raised similar doubts. At the same time, we thank you for the specific examples provided (such as the constraint of layout on connection type), which intuitively reveal the coupling relationship between attributes. Addressing your questions regarding "what is the actual cost" and "how much error will be introduced," we hereby make a clear clarification: **We acknowledge that the independence assumption is a first-order approximation made for computational feasibility. The actual cost of this assumption does not lead to the "incoherent questioning sequence" you worry about, but rather manifests as "conservativeness and redundancy" in strategy.** We will explain the impact brought by this trade-off from the following three dimensions:
> > > - **Necessary Trade-off in Computational Complexity.** Our state space contains over 35,000 elements. If we were to model the fully coupled relationships of all attributes according to reality, it would bring exponential computational overhead, rendering real-time inference infeasible from an engineering perspective. The independence assumption reduces computational complexity from exponential to linear; this is a necessary compromise to achieve system real-time response.
> > > - **Theoretical Cost: Conservativeness of Strategy.**  You asked what the "actual cost" is. Due to ignoring the dependencies between attributes, the uncertainty perceived by the system will be higher than the actual value. At the policy level, this causes the model to become "conservative": meaning it might ask some confirmatory questions that could have been inferred (e.g., after determining the layout, it still needs to confirm the connection type), thereby resulting in a slight increase in interaction turns, which is sacrificing a certain amount of efficiency.
> > > - **Practical Guarantee: Robustness brought by Hard Constraints.** Addressing your concern about "incoherent questioning," the robustness of this framework lies in our "hard-constraint update mechanism." Regardless of the prior assumption, once the user gives a definite answer, the posterior probability is forcibly collapsed to 1. This mechanism forcibly corrects the bias brought by the independence assumption. Therefore, the cost of violating this assumption is strictly limited to the level of interaction efficiency (possibly causing the dialogue to increase by a few turns), and will absolutely not lead to systemic failure or intention misjudgment.
> > >
> > >
> > >
> > > **6.Regarding Question 4: New Task Construction.** Addressing your inquiry on how to extend this framework to new domains, we have established a standardized, data-driven pipeline to ensure the feasibility of expansion. This process does not rely on extensive manual rules and mainly comprises three steps: First is Data Collection, i.e., aggregating the corpus or datasets of the target domain; Second is Schema Extraction, leveraging the powerful inductive capabilities of modern LLMs or rule-based parsers to extract structured attribute definitions from unstructured data; Finally is Prior Construction, establishing an empirical prior distribution by statistically analyzing the frequency of attribute values within the real data distribution. This pipeline ensures that the attribute space faithfully reflects the complexity of the domain and provides a reliable mathematical basis for the calculation of Information gain.

---

> > > > ### Author Response · Authors · 2025-11-21
> > > > **Official Comment by Authors**
> > > >
> > > > ## 4/4
> > > >
> > > > **7.Regarding Q5: Baseline Selection.** We thank you for raising this critical question. We cited works such as Geishauser et al. [6] and White et al. [7] to clarify the **theoretical origins of introducing information theory into dialogue systems**, rather than viewing them as direct engineering baselines. The reason lies in the mismatch of task nature: **existing works mostly serve discriminative "vertical convergence tasks," and their mathematical assumptions rely on full enumeration, which cannot handle the "open-ended construction task" we face**. To respond to your concerns, we conducted adaptive comparative experiments on UoT (NeurIPS 2024) and CQ-Gen (EMNLP 2023), the two most representative SOTA methods. The results confirm that when facing high-dimensional generation tasks, they suffer from computational infeasibility or goal misalignment. We will add the following explanation in the final version:
> > > >
> > > > - **Theoretical Mismatch: Mathematical Intractability of Enumeration Methods.** Existing works (such as White et al.'s "20 Questions" or Lee et al.'s QA disambiguation) aim to "lock" an answer from a closed set, relying on pool-based scoring (enumerating candidates to calculate IG). However, Nous deals with a construction task with a combinatorial explosion of the state space (>35,000 types). Using existing enumeration logic to calculate the entropy of a high-dimensional generation space during real-time interaction encounters the "curse of dimensionality." Therefore, specialized "retrieval" methods are difficult to transfer, while prompt-based methods are the only paradigm capable of crossing domain boundaries.
> > > >
> > > >
> > > > - **Supplementary Experiments:** To strengthen our argument, we forcibly adapted the above two types of methods to this task. **Since they cannot output the final summary text description (lacking the ability to generate standardized completions), we only compared interaction efficiency and total accumulated information**, adopting two experimental settings: (A) Zero-start and (B) Draft-start.
> > > >
> > > > - **UoT (NeurIPS 2024) [8]:** Represents the genre based on simulation planning. Since full enumeration is infeasible, we restricted it to simulate only 5 paths per turn. This directly led to inaccurate IG estimation, and the model could not select the most efficient questions, resulting in a significant increase in interaction turns (average 35.6 turns) and incomplete final information acquisition.
> > > >
> > > > - **CQ-Gen (EMNLP 2023) [9]:** Represents the genre based on simulation planning. Since full enumeration is infeasible, we restricted it to simulate only 5 paths per turn. This directly led to inaccurate IG estimation, and the model could not select the most efficient questions, resulting in a significant increase in interaction turns (average 35.6 turns) and incomplete final information acquisition.
> > > >
> > > >
> > > >   | Prompt Type | Method | Turns (↓) | Total IG (↑) |
> > > >   | :--- | :--- | :---: | :---: |
> > > >   | Prompt_Zero | Nous (OfG) | 17.3 | 126.44 |
> > > >   | | UoT | 33.6 | 38.38 |
> > > >   | | CQ-Gen | 7.1 | 9.72 |
> > > >   | Prompt_Draft | Nous (OfG) | 7.6 | 42.31 |
> > > >   | | UoT | 37.6 | 12.30 |
> > > >   | | CQ-Gen | 9.7 | 14.61 |
> > > >
> > > >
> > > > Finally, we would like to extend our sincerest gratitude again for your rigorous and profound review. We have incorporated all the revision suggestions you put forward in the revised version and marked them in blue. We sincerely hope that these detailed responses have clarified your key concerns. If you are satisfied with our response, we would be most grateful if you could help us raise the score. We are deeply grateful, and we wish you all the best and send your warmest regards.
> > > >
> > > >
> > > > [1] Budzianowski, Paweł, et al. “MultiWOZ - A Large - Scale Multi - Domain Wizard - of - Oz Dataset for Task - Oriented Dialogue Modelling.” EMNLP, 2018.
> > > >
> > > > [2] Ramadan, Osman, et al. “Large - Scale Multi - Domain Belief Tracking with Knowledge Sharing.” ACL, 2018.
> > > >
> > > > [3] Eric, Mihail, et al. “MultiWOZ 2.1: Multi - Domain Dialogue State Corrections and State Tracking Baselines.” arXiv preprint arXiv:1907.01669, 2019.
> > > >
> > > > [4] Zang, Xiaoxue, et al. “MultiWOZ 2.2: A Dialogue Dataset with Additional Annotation Corrections and State Tracking Baselines.” ACL 2020.
> > > >
> > > > [6] Geishauser, C., S. Hu, H. Lin, N. Lubis, M. Heck, S. Feng, C. van Niekerk, and M. Gašić. "What Does the User Want? Information Gain for Hierarchical Dialogue Policy Optimisation." ASRU, 2021.
> > > >
> > > > [7]White, Julia, et al. "Open-domain clarification question generation without question examples." arXiv, 19 Oct. 2021, arXiv:2110.09779.
> > > >
> > > > [8] Hu, Zhiyuan, et al. “Uncertainty of Thoughts: Uncertainty-Aware Planning Enhances Information Seeking in Large Language Models.” NeurIPS, 2024,
> > > >
> > > > [9] Lee, Dongryeol, Segwang Kim, Minwoo Lee, Hwanhee Lee, Joonsuk Park, Sang-Woo Lee, and Kyomin Jung. "Asking Clarification Questions to Handle Ambiguity in Open-Domain QA." EMNLP 2023.

---

### Official Review · Reviewer_afyP · 2025-11-01

**Soundness:** 3
**Presentation:** 3
**Contribution:** 3
**Rating:** 6
**Confidence:** 2

**Summary:**

This paper proposes a Socratic collaboration paradigm called Nous, which actively asks clarifying questions to address user intent ambiguity in high-dimensional structured tasks. The method formalizes conversational progress as information gain, equivalent to a reduction in Shannon entropy within the task attribute space, and utilizes this intrinsic signal to train the questioning strategy without requiring human preference labels. The approach introduces an automated simulation process that uses an oracle with knowledge of the true specifications to generate conversational data with preference rankings, and applies a group relative policy optimization method in an offline setting. Experiments on a scientific diagram generation task show that Nous outperforms SFT, DPO, and prompt-based GPT/Qwen baselines in both interaction efficiency and output quality.

**Strengths:**

Novel formulation of the collaborative task. This paper proposes a novel framework that uses principle-based uncertainty reduction process and defines the reward signal as information gain (entropy reduction) for intention expression gap problem in human-AI collaboration.

Proposes automated simulation pipeline to generate a large-scale, preference-based dataset for the challenging task of scientific diagram generation. considers practicality and scalability.

**Weaknesses:**

1. The information-theoretic framework relies on the simplifying assumption that task attributes are conditionally independent. This might not be true for many complex, real-world tasks where choices are coupled (e.g., a specific graph layout choice might constrain the types of available components). The paper does not sufficiently analyze the impact of violating this assumption.

2. The evaluation is conducted in a simulated environment using a template Oracle agent that provides structured responses. This setup does not capture the complexity of real human interaction, where user inputs may be ambiguous noisy, unstructured. It leaves a significant gap in demonstrating the system's robustness and utility in a real-world collaborative setting.

**Questions:**

Regarding the conditional independence assumption, could you elaborate on its potential impact in more complex domains? What would be the trade-offs of moving beyond the independence assumption?

Your work relies heavily on a simulated Oracle with templated responses. Could you provide more detail on the robustness of the semantic parser to noisy, non-templated, or ambiguous user inputs that are more representative of real human language? Even a small-scale human study involving real users interacting with Nous would strengthen the paper's claims.

---

> ### Author Response · Authors · 2025-11-21
> **Official Comment by Authors**
>
> ## 1/2
>
> We sincerely appreciate you for your insightful and detailed review comments. Your positive feedback is a great encouragement to our team, and we appreciate your recognition of our work's strengths, especially the novel conceptualization of the uncertainty reduction framework and the practicality of our automated simulation pipeline. Furthermore, we have carefully addressed every question you raised and incorporated your feedback into the revision of the paper. These changes are highlighted in blue in the revised version for your reference.
>
> **1.Regarding Weakness 1 & Question 1: Conditional Independence Assumption**, we thank you very much for pointing this out. You and Reviewer BJFA raised the same doubt, and we hereby provide a consistent clarification. We fully agree with your point that "various choices are correlated." Since we neglected to analyze this issue in the text, which caused a misunderstanding, we hereby supplement: We fully acknowledge that this is **a first-order approximation made for computational feasibility. Although ignoring the correlations between attributes causes the system to overestimate entropy, we employed a hard-constraint update mechanism to offset this impact from an engineering perspective**. The trade-off analysis for abandoning the independence assumption is as follows:
>
> - **Necessary Trade-off in Computational Complexity.** Our state space contains over 35,000 elements. If we were to model the fully coupled relationships of all attributes, it would bring **exponential computational overhead**, rendering real-time inference infeasible. The independence assumption reduces the complexity to linear, which is a necessary compromise for engineering implementation.
> - **Theoretical Impact: Entropy Overestimation and Suboptimal Strategy.** Ignoring the "Mutual Information" between attributes means the uncertainty perceived by the system is higher than the actual value. This causes the policy to be biased towards being "conservative," meaning the agent might ask some redundant questions that could have been inferred (e.g., still asking about the highly correlated "color scheme" after determining the "style"), thereby resulting in a slight increase in interaction turns.
> - **Practical Guarantee:** The robustness of this framework lies in our "practical guarantee." The "hard-constraint update mechanism" we adopted ensures that the system can recover from this bias. Once the user gives a definite answer, the posterior probability is forcibly collapsed to 1. This ensures that the cost of violating the assumption is strictly limited to interaction efficiency (possibly causing the dialogue to increase by a few turns), and will **absolutely not lead to intention misjudgment or systemic failure**.

---

> > ### Author Response · Authors · 2025-11-21
> > **Official Comment by Authors**
> >
> > ## 2/2
> >
> > **2.Regarding Weakness 2 and Question 2: Simulators and Real Users**, this is another critical issue you raised, and it is also the point we most wish to clarify. We fully agree that simulators cannot completely capture the complexity of human interaction. To this end, we adopted a combination of "Simulator Benchmark + Real User Experiment Verification + Noise Resistance Ablation Study." Our experiments show that **Nous not only performs excellently in controlled simulations** but also, **due to the high information density of human language, its actual efficiency in real human interactions often outperforms the simulator.** Furthermore, our semantic parser demonstrated extremely **strong robustness against ambiguous and noisy inputs**.
> > - **Necessity of the Simulator (Standardized Benchmark).** Human interaction possesses high variability (e.g., humans tend to reveal more information; for instance, when Nous only asks a question to determine the overall style, human users often answer with information regarding style, color, layout, etc., all at once). To ensure experimental reproducibility and variable control (fair comparison of different algorithms), using a behaviorally controllable simulator is a necessary prerequisite for building a standardized benchmark.
> > - **Real User Experiment Verification (Existing and Supplementary).**
> >   - **Existing Experiments (Section 4.3):** We already included a user study involving 3 PhD students (100 trials) in the paper. Results show that despite the ambiguity in human expression, due to higher information density (single responses containing more information), interaction turns were actually fewer than the simulator (18.7 and 20.3), and the final quality was comparable.
> >   - **Supplementary Experiments:** To further respond to your concerns, we invited 7 participants to conduct more in-depth testing, divided into two groups: (A) Zero-start and (B) Draft-start. The results (see Table 2) reaffirm our conclusion: Real human interaction is highly consistent with the simulator in terms of quality, while in the practical scenario introducing "Draft Prompts," interaction turns were significantly shortened to 7.6 turns, fully proving the practicality of the framework.
> >
> >
> > - **Stability of the Semantic Parser.**  Addressing concerns about ambiguous, unstructured inputs, we conducted a dedicated ablation study in Appendix C (when constructing data, **we used two settings: providing incomplete information and introducing ambiguous noise information**, resulting in Nous-Vague and Nous-Noisy). Experimental results indicate that even if the training data contains a large amount of ambiguous or irrelevant information, model performance did not degrade. It learned the core strategy of identifying high-information-gain questions from ambiguous and noisy dialogues. This demonstrates the stability of the semantic parser when facing vague content and noisy signals.
> >
> >
> >   | Method | Turns (↓) | Total IG (↑) | Score (↑) | Human Win | Human Rate* | Human W/T | GPT-5 Win | GPT-5 Rate* | GPT-5 W/T |
> >   | :--- | :---: | :---: | :---: | :---: | :---: | :---: | :---: | :---: | :---: |
> >   | Expert Oracle | 20.3 | 120.53 | **0.76** | 0.41 | 0.50 | 0.59 | 0.33 | 0.49 | **0.65** |
> >   | Novice Oracle | 24.1 | 122.47 | 0.74 | **0.42** | 0.49 | 0.57 | **0.38** | 0.51 | 0.64 |
> >   | Human User (Zero) | 17.3 | **126.44** | 0.75 | 0.40 | **0.51** | **0.61** | 0.38 | 0.49 | 0.61 |
> >   | Human User (Draft) | **7.6** | 42.31 | 0.74 | 0.42 | 0.50 | 0.59 | 0.37 | **0.51** | 0.64 |
> >
> >
> >
> > We sincerely thank you again for your time and the comprehensive review. Your feedback has been invaluable in strengthening our manuscript.We have incorporated all the revision suggestions you put forward in the revised version and marked them in blue. We hope that our responses and revisions have adequately addressed your concerns. If you are satisfied with our clarifications, we would kindly ask you to consider raising your score. We wish you all the best and send our warmest regards.

---

### Official Review · Reviewer_WZwJ · 2025-11-10

**Soundness:** 3
**Presentation:** 3
**Contribution:** 2
**Rating:** 4
**Confidence:** 4

**Summary:**

This paper studies the goal-oriented dialogue generation for scientific diagram generation. The author proposes to leverage the information gain from dialogue as an intrinsic reward signal and leverage the standard post-training methods, i.e., offline GRPO (OfG), to post-train the LLM to efficiently and effectively capture the user intent for scientific diagram generation, and then use the user answers and other text2image (T2I) models to generate the scientific diagram. To enable the training, the author proposes an automatic data generation pipeline to curate the data from the public academic research papers. Experiments are conducted on the generated datasets under different variants of post-training methods and the choice of LLM, where the OfG is better than other baselines.

**Strengths:**

1. The studied problem lies in the interaction between HCI and LLM, which is quite interesting.

2. The idea of leveraging the classic information theory to measure the uncertainty over the QA rounds to understand the user intent is standard and sound. The usage of offline GRPO shows effectiveness when compared with other baselines.

**Weaknesses:**

1. The practicality of the problem setup. This paper considers the problem setup in which the image, e.g., the scientific diagram, is generated after the LLM has fully resolved the user's intent in text. However, asking too much information may be potentially redundant and unnecessary. In reality, it is more common to see the use cases that the image is generating alongside the QA, and the user can also consider the intermediate generation results to determine whether it is still necessary to proceed with the QA. How will the proposed method work under such a practical setup?

2. The comparison methods are limited. There are no existing methods in scientific diagram generation via LLM that are compared in Section 4, while only a simple baseline, e.g., zero-shot inference on GPT and Qwen, is presented. This may not be sufficient to evaluate the significance of the proposed method comprehensively.

**Questions:**

Please refer to the Weaknesses section.

---

> ### Author Response · Authors · 2025-11-21
> **Official Comment by Authors**
>
> ## 1/2
>
> We sincerely appreciate your constructive feedback. We are grateful for your recognition of our work's strengths, particularly regarding the importance of the studied problem and the effectiveness of our proposed method. We have carefully addressed each of your concerns one by one and updated the manuscript to incorporate these clarifications. For your convenience in reviewing, the revised parts have been marked in blue.
>
> **1.Regarding Weakness 1: Practicality and Intermediate Generation**, we thank you for your insight regarding the practical scenario of "generation alongside feedback." We agree that this is a common interaction mode; however, our proposed Nous is intended to serve as a **pre-generation reasoning module**. The purpose is to **eliminate fundamental ambiguities through low-cost textual interaction before the expensive rendering process begins**. Our method complements existing practical scenarios in the following three aspects:
>
> - **Ambiguity in Description and Reducing Cognitive Load.** It is difficult for users to perfectly **express high-dimensional visual intents via a single textual input**. Vague inputs force the model to "guess," leading users into an inefficient "generate-modify" trial-and-error loop. We argue that confirming key constraints (e.g., "whether it is a directed graph") textually before generation imposes a cognitive load far lower than performing visual corrections after generating an erroneous complex image.
> - **Context Window Efficiency.** This repeated "trial-and-error loop" is technically expensive. Repeatedly inserting huge image tokens in multi-turn dialogues will rapidly exhaust the context window, causing the model to suffer from "forgetting" or "hallucination." Nous chooses to resolve uncertainty in a lightweight textual space, fundamentally avoiding this vicious cycle.
> - **Efficiency in Actual Scenarios.** Regarding your concern about "excessive inquiries," our adoption of the "zero-start" setting in the main experiments was to strictly control variables. In real-world applications, users usually provide a "Draft Prompt" containing partial information. Our supplementary experiments show that in this scenario (Human User_Draft), the model utilizes prior information to significantly reduce interaction turns, and there is no issue of redundant questioning.
>
>
>   | Method | Turns (↓) | Total IG (↑) | Score (↑) | Human Win | Human Rate* | Human W/T | GPT-5 Win | GPT-5 Rate* | GPT-5 W/T |
>   | :--- | :---: | :---: | :---: | :---: | :---: | :---: | :---: | :---: | :---: |
>   | Expert Oracle | 20.3 | 120.53 | **0.76** | 0.41 | 0.50 | 0.59 | 0.33 | 0.49 | **0.65** |
>   | Novice Oracle | 24.1 | 122.47 | 0.74 | **0.42** | 0.49 | 0.57 | **0.38** | 0.51 | 0.64 |
>   | Human User (Zero) | 17.3 | **126.44** | 0.75 | 0.40 | **0.51** | **0.61** | 0.38 | 0.49 | 0.61 |
>   | Human User (Draft) | **7.6** | 42.31 | 0.74 | 0.42 | 0.50 | 0.59 | 0.37 | **0.51** | 0.64 |
>
> [1]Han, Yucheng, et al. "ChartLlama: A Multimodal LLM for Chart Understanding and Generation." arXiv, 27 Nov. 2023, arXiv:2311.16483.
>
> [2] Geishauser, C., S. Hu, H. Lin, N. Lubis, M. Heck, S. Feng, C. van Niekerk, and M. Gašić. "What Does the User Want? Information Gain for Hierarchical Dialogue Policy Optimisation." 2021 IEEE Automatic Speech Recognition and Understanding Workshop (ASRU), 2021.
>
> [3] White, Julia, et al. "Open-domain clarification question generation without question examples." arXiv, 19 Oct. 2021, arXiv:2110.09779.
>
> [4]Hu, Zhiyuan, et al. “Uncertainty of Thoughts: Uncertainty-Aware Planning Enhances Information Seeking in Large Language Models.” Advances in Neural Information Processing Systems (NeurIPS), 2024,
>
> [5] Lee, Dongryeol, Segwang Kim, Minwoo Lee, Hwanhee Lee, Joonsuk Park, Sang-Woo Lee, and Kyomin Jung. "Asking Clarification Questions to Handle Ambiguity in Open-Domain QA." Findings of the Association for Computational Linguistics: EMNLP 2023.

---

> > ### Author Response · Authors · 2025-11-21
> > **Official Comment by Authors**
> >
> > ## 2/2
> >
> > **2.Regarding Weakness 2: Insufficient Baseline Comparison**, we truly appreciate your valuable feedback regarding baseline selection, and we will make every effort to eliminate your doubts or misunderstandings. Addressing your query regarding "existing methods for scientific diagram generation via LLMs" and the "simple baselines," we provide the following clarifications: The core of our work is not "passive rendering," but **"active intention clarification" prior to generation**. Existing generation works cannot bridge the intention gap,**while existing clarification works are mathematically difficult to handle high-dimensional construction tasks. Therefore, Socratic questioning with general-purpose LLMs is currently the only feasible and fair comparison target.** To strengthen our argument, we introduced UoT (NeurIPS 2024) and CQ-Gen (EMNLP 2023) for supplementary comparison, and the results confirm that they have severe bottlenecks in efficiency and information acquisition. To this end, we make the following clarifications:
> > - **Difference in Task Positioning: Pre-clarification and Back-end Rendering** We speculate that the "existing generation methods" you mentioned refer to Prompt optimization or code generation tools (such as ChartLlama [1], etc.). Nous is not a "rendering tool," but its "pre-generation reasoning module." Existing research typically assumes the input is perfect, focusing on the mapping from text to code; whereas we focus on addressing the neglected "pre-collaboration" intention clarification bottleneck before the "trial-and-error loop." Therefore, directly comparing rendering capabilities cannot effectively evaluate the quality of "inquiry strategies."
> > - **Theoretical Mismatch: Why Other Clarification Methods Were Not Compared**  We cited works such as Geishauser et al. [2] and White et al. [3] to acknowledge their theoretical origins. The reason for not using them as baselines lies in mathematical intractability: These methods mostly serve "vertical convergence tasks" (such as "20 Questions" guessing games or QA disambiguation), relying on full enumeration of the candidate pool to calculate Information Gain. Our task is "open-ended construction," where the state space suffers from combinatorial explosion (exceeding 35,000 combinations). Conducting full enumeration during real-time interaction would encounter the "curse of dimensionality." Therefore, those methods dedicated to "retrieval" are difficult to directly transfer, while prompt-based (GPT/Qwen) methods are currently the only general paradigm capable of crossing domains and handling open-ended inquiry.
> > - **Supplementary Experiments:**  To respond to your concerns, we applied two representative methods to this task. Since they **cannot output the final summary text description** (lacking the ability to generate standardized completions), **we only compared interaction efficiency and total accumulated information**. We adopted two experimental settings divided into (A) Zero-start and (B) Draft-start groups:
> >   - UoT (NeurIPS 2024) [4]: Based on simulation planning. Because full enumeration is infeasible, we restricted it to sparse sampling (N=5). Results show that the model failed to ask efficient questions due to inaccurate valuation, leading to a surge in interaction turns (average 35.6 turns), and low information acquisition efficiency.
> >   - CQ-Gen (EMNLP 2023) [5]: Based on semantic prompting. The model tends to ask about high-level semantics such as "usage" and cannot track structural parameters. Although the dialogue is shorter, its total accumulated information is extremely low (average only 12.2), proving its failure to complete the construction of structural information.
> >
> >
> >   | Prompt Type | Method | Turns (↓) | Total IG (↑) |
> >   | :--- | :--- | :---: | :---: |
> >   | Prompt_Zero | Nous (OfG) | 17.3 | 126.44 |
> >   | | UoT | 33.6 | 38.38 |
> >   | | CQ-Gen | 7.1 | 9.72 |
> >   | Prompt_Draft | Nous (OfG) | 7.6 | 42.31 |
> >   | | UoT | 37.6 | 12.30 |
> >   | | CQ-Gen | 9.7 | 14.61 |
> >
> > Finally, we would like to thank you again for your valuable opinions and pertinent suggestions. We have incorporated all the revision suggestions you put forward in the revised version and marked them in blue. We hope that our clarifications and the revisions in the manuscript have adequately addressed the core issues you raised. If you are satisfied with our response, we would greatly appreciate it if you could consider raising your score. We wish you all the best.

---

### Author Response · Authors · 2025-11-27
**Kind reminder of the discussion**

We sincerely thank all reviewers and ACs for your constructive and detailed reviews. We have addressed the comments point-by-point, and revised our paper according to the suggestions, denoted in blue in our revised paper.  We appreciate Reviewers QBG3 and afyP for your positive scores, and we also appreciate the constructive questions raised by reviewers WZwJ and BJFA. We would like to know if our response has addressed your concerns and questions. If you have any further concerns or suggestions for the paper or our rebuttal, please let us know. We would be happy to engage in further discussion and manuscript improvement.

---

### Author Response · Authors · 2025-11-30
**Summary of Rebuttal and Revisions for AC Review**

We sincerely appreciate the ICLR ACs for participating in this review process and their efforts to uphold community interests and review fairness. Below is a concise summary of all comments from the four reviewers for your reference. For both common and specific issues raised, we have made detailed revisions to the main text and Appendix (all highlighted in blue).

**Common Issues:**

  - **[Additional Experiments] Issue 1:** Performance in real-world scenarios (WZwj: weakness1; afyP: weakness2 and question2; BJFA: weakness2 and question3; QBG3: weakness1 and question1). Initially, we recruited 3 PhD students for 100 tests each to simulate real-world scenarios, but reviewers noted this evidence was weak. To address this, during the rebuttal period, we recruited an additional 7 participants (**10 total**) to conduct **2,000** real-world tests across two settings. The results are **highly consistent** with the initial version, significantly strengthening the robustness of our conclusions.

  - **[Theoretical Analysis] Issue 2:**  Independence Assumption (afyP: weakness1 and Question1; BJFA: weakness5 and question1). We analyzed this in Appendix of the initial version, but **insufficient detail** led to misunderstandings by the reviewers. Therefore, in the rebuttal phase, we provided detailed explanations covering three aspects: computational complexity trade-offs, theoretical costs, and practical effects. We have also added detailed supplementary explanations in the paper.

  - **[Clarification & Additional Experiments] Issue 3:** Comparative Methods (WZwj: weakness2; BJFA: question5). Reviewers noted we cited many works based on information gain but did not include them for comparison. We clarified that citing these works **aimed to elucidate theoretical roots of information theory**, not to treat them as baselines. Due to **fundamental task differences**, current works do not support direct comparison in terms of vertical tasks and mathematical foundations. To address the reviewers' expectations, we introduced comparative experiments with two related works, disregarding the differences in task completion. Results show our system still demonstrates an absolute advantage.

**Specific Issues:**
  - **[Clarification] Issue 1:** Essential Difference from TOD (BJFA: weakness1). We suspect insufficient analysis in the initial Related Work section (main text and Appendix) caused ambiguity in task definition and reviewer misunderstanding. Therefore, we significantly expanded the relevant discussions in the main text and Appendix, explicitly delineating the fundamental differences from traditional TOD via two dimensions: "task definition" and "interaction goals."
  - **[Clarification] Issue 2:** Rationality of Unweighted (BJFA: weakness3 and question2). We believe this reflects a core theoretical misunderstanding; the entropy-based reward mechanism is one of our core contributions, and **we demonstrated our scheme’s superiority via ablation study comparisons.** To eliminate the reviewer's doubts, we provided a detailed clarification in our response from both theoretical and experimental perspectives.

  - **[Clarification] Issue 3:** Generalization Evidence and Applicability Boundaries (BJFA: weakness4 and question6).The reviewer argued that conducting only one generalization experiment is not persuasive. We emphasize the core goal is verifying "**mechanism effectiveness**" rather than exhaustive scenario testing, and we have observed this from generalization experiments. To address concerns, we added discussions on the high-baseline effect and applicability boundaries in the response and paper.
  - **[Clarification] Issue 4:** New Task Construction (BJFA: question4). Regarding this issue, we have provided a clear and detailed elaboration in the data construction sections of the main text and Appendix. To resolve the reviewer's doubt, we further clarified this in our response and refined relevant descriptions in the paper to avoid future oversight.

  - **[Theoretical Analysis] Issue 5:** Validity of Metrics (QBG3: weakness2). The insufficient analysis in our initial version caused doubts for the reviewer. We provided supplementary explanations for the relevant metrics in our response and updated the manuscript accordingly.
  - **[Revision] Issue 6:** Terminology Issues (QBG3: question2). Our use of the term "World Model" could indeed cause misunderstandings. We fully adopted the reviewer's suggestion to change it to "Empirical Prior" and modified the content throughout the text.


To date, we have systematically responded to all reviewers with substantive revisions, addressing every identified weakness and question, and resolving all misunderstandings. Regrettably, we did not receive further feedback from any reviewer during the rebuttal. We earnestly request the AC to fully consider our detailed rebuttal efforts and proactive clarifications in the evaluation.

Thank you again for your efforts in this review process

---

### Meta-Review · Area_Chair_qBUK · 2026-01-07

**Summary:**

This paper proposes a mechanism whereby an agent proactively requests the user to reduce uncertainty about user intent. This is done by using information gain as a reward signal during post-training. The approach is evaluated on a scientifict diagram generation task.

The idea is quite interesting but is too focused on the specific diagram-generation task, whereas the authors seem to argue for a general purpose method, as seen in the title and most of the abstract. Because of this, and due to the reviewer concerns listed below, I do not believe this work is ready for publication.

To strengthen this work I would suggesting either running evaluations on more tasks (as suggested by a number of reviewers) or doubling down on this diagram-generation task, and make it clear that the submission is focused (and limited) on this specific problem. The novel writing task is a nice start, but the results do not appear to be convincing enough. Finally, the authors should also provide confidence intervals and/or error bounds before claiming statistical significance.

Another suggestion would be to propose this diagram-generation task (which appears to be rather novel) as a new challenge/benchmark; under this framing, the approach proposed could be included as an initial baseline.

**Reviewer Concerns:**

## WZwJ
- W1 (practicality of the problem setup): reviewer suggests that the generated image should be part of the iteration loop. The reviewers argue that their method is a “pre-generation reasoning module”. If the work were exclusively focused on diagram-generation, this would be fine; however, as a general purpose solution, it would have been more convincing to provide text-only experiments where there isn’t a “post-generation” component which is critical to evaluation.
- The reviewer (rightfully) suggests that the methods compared against are insufficient. The authors respond by saying that “we focus on addressing the neglected "pre-collaboration" intention clarification bottleneck before the "trial-and-error loop." Therefore, directly comparing rendering capabilities cannot effectively evaluate the quality of "inquiry strategies."” This once again suggests that the manuscript as is is too focused on the diagram-generation task.

## afyP
- W1 (conditional independence assumption of task variables). The authors respond that it is “a necessary compromise for engineering implementation”, once again suggesting that the work is limited to diagram generation.
- W2 (oracle is too idealistic). Here the authors provided a reasonable rebuttal, but once again rather focused on the specific task at hand.

## BJFA
- W1 (relationship to traditional task-oriented dialogue is unclear). The authors ask, relative to existing methods such as MultiWOZ: “what is fundamentally different here beyond applying it to diagram generation instead of restaurant booking?”. The author response is once again too focused on this specific problem (e.g. “The target diagram needs to be "created" from scratch”).
- W2 (All evaluation occurs in simulation with idealized conditions and effectiveness of real-user experiments). The authors respond somewhat ambiguously: “In preexperiments, humans tended to "overshare" (e.g., answering multiple attributes at once)”.
  - Further, the authors say “In the paper, 3 PhD students completed 300 interaction logs, ensuring statistical significance”, but the authors do not provide any confidence intervals or error bounds, which calls into question their claims of statistical significance.
- W3 (The entropy objective lacks justification for this specific task.). The authors respond adequately _for this specific task_, but not for a general purpose solution.
- W4 (Generalization evidence is weak.) Although the authors do provide a novel writing task, the evidence is not as strong as it is for the diagram generation. The applicability boundaries (added during rebuttal) do help clarify on what tasks the paper is focused on, but she be brough more up-front.
- W5 (The independence assumption oversimplifies the problem structure). The answers are once again quite focused on diagram-generation.
- Q4 (New Task Construction). The authors provide a reasonble approach on how to extend the framework to new domains, so I would enourage the authors to follow this to provide concrete evidence of their proposal.
- Q5 (Why are established clarification dialogue methods not used as baselines?). The authors posit that “existing works mostly serve discriminative "vertical convergence tasks," and their mathematical assumptions rely on full enumeration, which cannot handle the "open-ended construction task" we face”. It seems like their method should be able to be adapted to "vertical convergence tasks"; if they can’t, then it calls into question the generalizability of the approach.
- Q6 (Applicability boundaries). The authors addressed this adequately with the text added to the revision.

## QBG3
- W1 (The hard-constraint update assumes a perfect parsing and unambiguous answer and according to which the paper attains using templated Oracle responses). While the authors provided a lengthy (and reasonble) response, it is focused solely on the diagram-generation task.
- W2 (Validity of VisPainter Metrics). The authors provided a reasonable response to this question.

**Reviewer Scores:**

- **WZwJ:** Currently at 4, unlikely to increase.
- **afyP:** Currently at 6, unlikely to increase their score.
- **BJFA:** Currently at 2, may have increased score slightly but unlikely to go above a 4.
- **QBG3:** Currently at 8, unlikely to increase their score.

---

### Decision · Program_Chairs · 2026-01-26

Reject